# A GWAS in Latin Americans highlights the convergent evolution of lighter skin pigmentation in Eurasia

Kaustubh Adhikari et al.[#]

We report a genome-wide association scan in >6,000 Latin Americans for pigmentation of skin and eyes. We found eighteen signals of association at twelve genomic regions. These include one novel locus for skin pigmentation (in 10q26) and three novel loci for eye pigmentation (in 1q32, 20q13 and 22q12). We demonstrate the presence of multiple independent signals of association in the 11q14 and 15q13 regions (comprising the *GRM5/TYR* and *HERC2/OCA2* genes, respectively) and several epistatic interactions among independently associated alleles. Strongest association with skin pigmentation at 19p13 was observed for an Y182H missense variant (common only in East Asians and Native Americans) in *MFSD12*, a gene recently associated with skin pigmentation in Africans. We show that the frequency of the derived allele at Y182H is significantly correlated with lower solar radiation intensity in East Asia and infer that *MFSD12* was under selection in East Asians, probably after their split from Europeans.

Hundreds of genes involved in pigmentation have been identified in animal models (http://www.espcr.org/micemut/) and mutations at some of these have been shown to cause rare human pigmentation disorders[1]. Extensive association analyses have robustly identified polymorphisms at tens of pigmentation genes impacting on variation of skin, eye or hair color in humans[2,3], the great majority of these variants have been identified in European-derived populations. Recent analyses of non-European populations have suggested the existence of additional pigmentation variants, emphasizing the importance of a wider population characterization in order to obtain a fuller picture of the genetic architecture of pigmentation variation in humans[4,5].

Since Darwin's original proposal, it has been suggested that the evolution of pigmentation in humans (and other organisms) could have been shaped by some form of selection[6,7]. In particular, the observation of a decrease in human skin pigmentation at increasing distance from the Equator has been interpreted as resulting from an adaptation to lower levels of ultraviolet radiation, consistent with the tanning response being a physiological skin-protection mechanism[8]. As a corollary, it has been suggested that variation in eye and hair color in Western Eurasians could represent a by-product of natural selection on skin pigmentation. Alternatively, it has been proposed that variation in human pigmentation could have been affected by sexual selection, or a form of frequency-dependent selection, as appears to be the case in many other animals[6,9].

In agreement with these evolutionary scenarios, analyses of patterns of human genome diversity have found signals of selection at certain pigmentation loci[5,10,11]. Interestingly, these signals were observed to only partially overlap between Europeans and East Asians, leading to the suggestion that variation in skin pigmentation could have evolved somewhat independently in Western and Eastern Eurasia[1,12]. Among the genomic regions affecting pigmentation in Europeans, variants in OCA2 and MC1R restricted to East Asia have been shown to impact on skin pigmentation in populations from this geographical area[13,14]. The fact that different alleles at these two genes impact on skin pigmentation in Western and Eastern Eurasia agrees with the evolutionary convergence of lighter skin color in these two regions[1,15]. Thus, further analyses of pigmentation in East Asian-derived populations are of special interest for examining the genetic architecture and evolution of lighter skin pigmentation in Eurasia.

To this end, here we report a genome-wide association study (GWAS) of pigmentation in over 6000 Latin Americans, most with high Native American ancestry[16]. It is well established that Native Americans are closely related to East Asians, the initial settlement of the New World starting some 15,000 years ago, through migration from Eastern Siberia into North America[17]. We identified four novel associated regions involving skin or eye pigmentation. Follow-up analyses conditioned on six well-established pigmentation variants (and explaining a large proportion of phenotypic variation in our sample) increase the strength of association for the other associated loci, and identified one additional locus known to impact on skin pigmentation. Furthermore, we detected an association signal for skin pigmentation within the MFSD12 gene, which is strongest for an Y182H amino-acid variant that is common only in East Asians and Native Americans. Other variants of MFSD12 have recently been shown to impact on skin pigmentation in Africans[5]. We find that the MFSD12 region shows significant evidence of selection in East Asians (dated after their split from Europeans) and that the frequency of the Y182H variant correlates with the intensity of solar radiation. We also explored the genetic architecture of pigmentation in Latin Americans, and found multiple independent signals of association at the 11q14 and 15q23 regions (overlapping GRM5/TYR and HERC2/OCA2), as well as signals of epistatic interactions among independently associated alleles. Overall, our findings highlight the complex genetic architecture of pigmentation phenotypes in Latin Americans, and support the view that, in modern humans, lighter skin pigmentation has evolved independently at least twice in Eurasia, possibly as an adaptation to geographic variation in solar radiation exposure.

## Results

**Pigmentation features examined**. Our study sample is part of the CANDELA cohort ascertained in five Latin American countries (Brazil, Colombia, Chile, Mexico and Peru; Supplementary Table 1)[16]. Information on skin, hair and eye (iris) pigmentation (Figs. 1 and 2) was obtained for 6357 individuals. Skin pigmentation, measured using reflectometry by the melanin index (MI), showed extensive variation. The MI ranged from 20 to 65 (mean = 34.98 and SD = 5.34). The lightest mean pigmentation was observed in Brazil (32.04) and the darkest mean pigmentation in Mexico (36.32) (Fig. 1a). We have previously reported genome-wide association analyses of categorical hair color in the CANDELA sample[18]. The most prevalent colors were black and brown, which account for ~80% of this sample. These were also the most prevalent categories across countries, except in Brazil where ~50% of individuals had dark blond/light brown or blond hair (Fig. 1b). Eye color was classified into 5 ordinal categories (1-blue/gray, 2-honey, 3-green, 4-light brown, 5-dark brown/black) by direct observation of the volunteers. The most common categories were dark brown/black and light brown, comprising ~85% of the sample (Fig. 1c). The lighter eye color categories (blue/gray and green) were more common in Brazil (~40%) than in in the other countries (≤10%).

In addition to eye color categories, we obtained quantitative variables related to perceived eye color from the analysis of digital photographs, using the HCL color space (hue, chroma, lightness) (Fig. 2 and Supplementary Figure 1–3). Hue (H) measures variation in color tone, whereas chroma (C) and lightness (L) measure saturation and brightness, respectively (Fig. 2a, b). The frequency distributions of these variables are shown in Supplementary Figure 4. In contrast to the eye color categories, these quantitative color variables capture variation not only in the blue/gray to brown spectrum (mainly captured by H and L), but also variation within the brown spectrum (mainly captured by C) (Fig. 2c, d): while individuals with the highest L values exhibited mainly blue/gray eyes, individuals with the highest C values exhibited eye colors with the lightest shades of brown (i.e., light brown or honey, Fig. 2c). As H is a circular variable, it was standardized and converted to cos(H) before testing for association (see Methods). In what follows we contrast results for all the pigmentation phenotypes examined in the CANDELA individuals.

All the pigmentation phenotypes examined are significantly (P values < 0.001) and positively correlated (Supplementary Table 2A). Strongest correlation was observed between hair and categorical eye color (r = 0.50), while there is lower correlation of these two traits with skin pigmentation (r = 0.30 and r = 0.31, respectively). Lighter pigmentation of hair, skin and eyes is also significantly (P values < 0.001) correlated with the genetic estimates of European ancestry (r ranging between 0.31 and 0.39, Supplementary Table 2B). Categorical eye color was strongly correlated with the L digital eye color variable (r = −0.78), but moderately correlated with cos(H) and almost uncorrelated with C (r of 0.40 and −0.08, respectively), highlighting the considerable amount of variation in the quantitative variables not captured by the eye color categories.

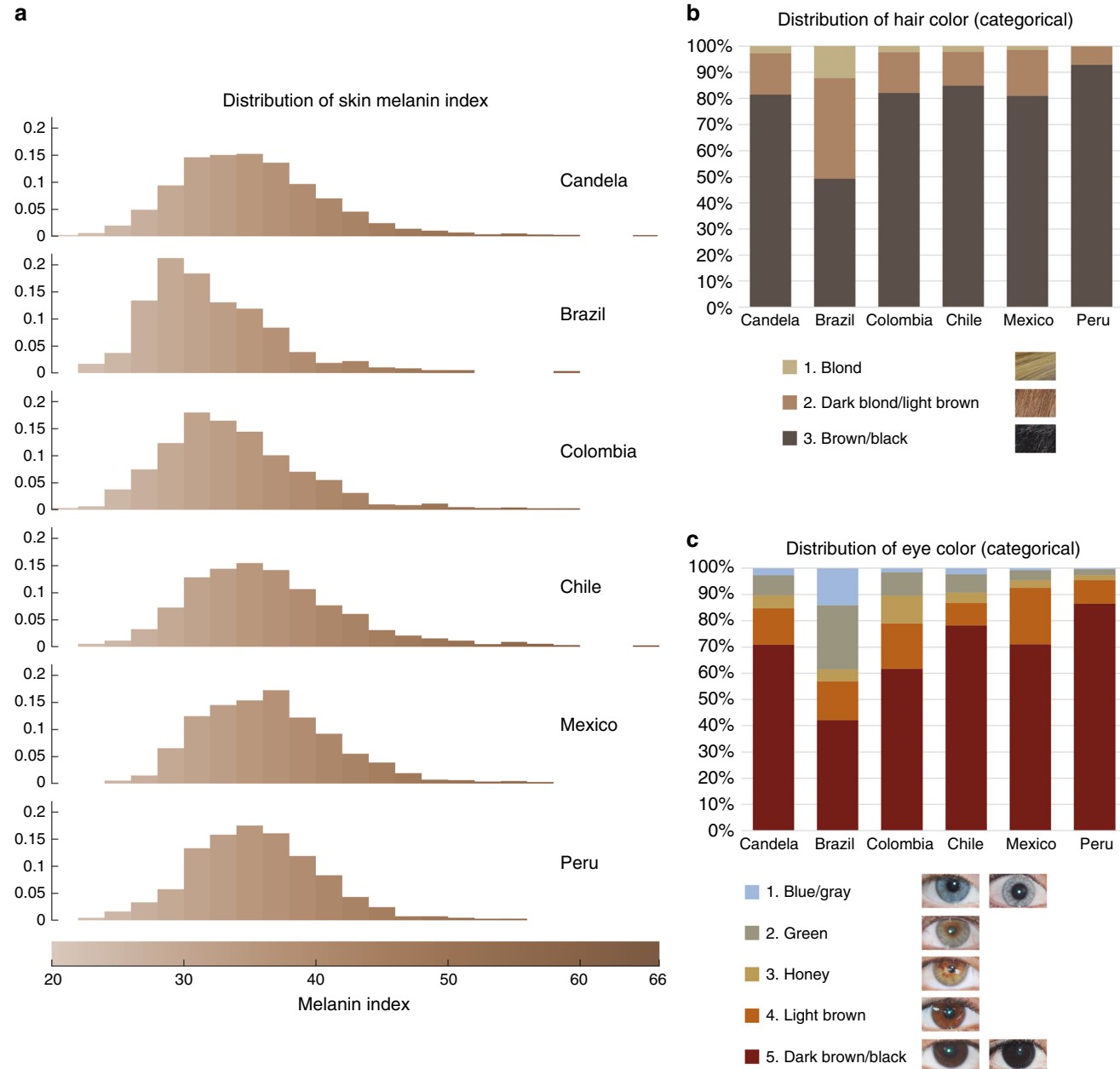

**Fig. 1** Distribution of skin, hair and eye pigmentation in the CANDELA sample. **a** Frequency distribution of skin melanin index (MI). Histograms are shown for the full CANDELA sample and for each country sample separately. To facilitate relating MI values to skin color, the MI values (x-axis) were converted to approximate RGB values (scale at the bottom, Supplementary Figure 16). **b** Stacked bar plots showing the frequency (percent) of the three hair color categories. Bar colors correspond approximately to the sample images for each category shown at the bottom (with the ordinal numbering used in the association analyses shown next to each category). **c** Stacked bar plots showing the frequency (percent) of eye color categories. Bar colors correspond approximately to the sample images of eyes as shown at the bottom (with the ordinal numbering used in the association analyses shown next to each category). Categories 1 and 5 are composite categories, respectively of blue/gray and dark brown/black and examples of each of the sub-type are shown

Individuals were genotyped on Illumina Omni Express BeadChip. After quality control, we retained 674,971 single-nucleotide polymorphisms (SNPs) and 6236 individuals for the genetic analyses. Average continental admixture proportions in these individuals were estimated as: 48% European, 46% Native American and 6% African (Supplementary Figure 5). Based on a kinship matrix obtained from the SNP data[19], we estimated a narrow-sense heritability for skin color of 0.85 (SE 0.05) and of 1 (SE 0.05) for both hair and eye color. Similarly, quantitative eye color variables showed high heritability estimates (between 0.79 and 1.00, SE 0.06) (Supplementary Table 3). High heritabilities

for pigmentation traits have also been estimated from family data[20,21].

**Association analyses**. The primary genome-wide association tests (Table 1) (using 8,896,142 genotyped and imputed SNPs) were performed using multivariate linear regression, as implemented in PLINK v1.9[22]. We used an additive genetic model adjusting for age, sex and the first six principal components (PCs; Supplementary Figure 6A) obtained from genome-wide SNP data. Following up the primary GWAS results, and to account for phenotypic variation explained by known pigmentation loci, we

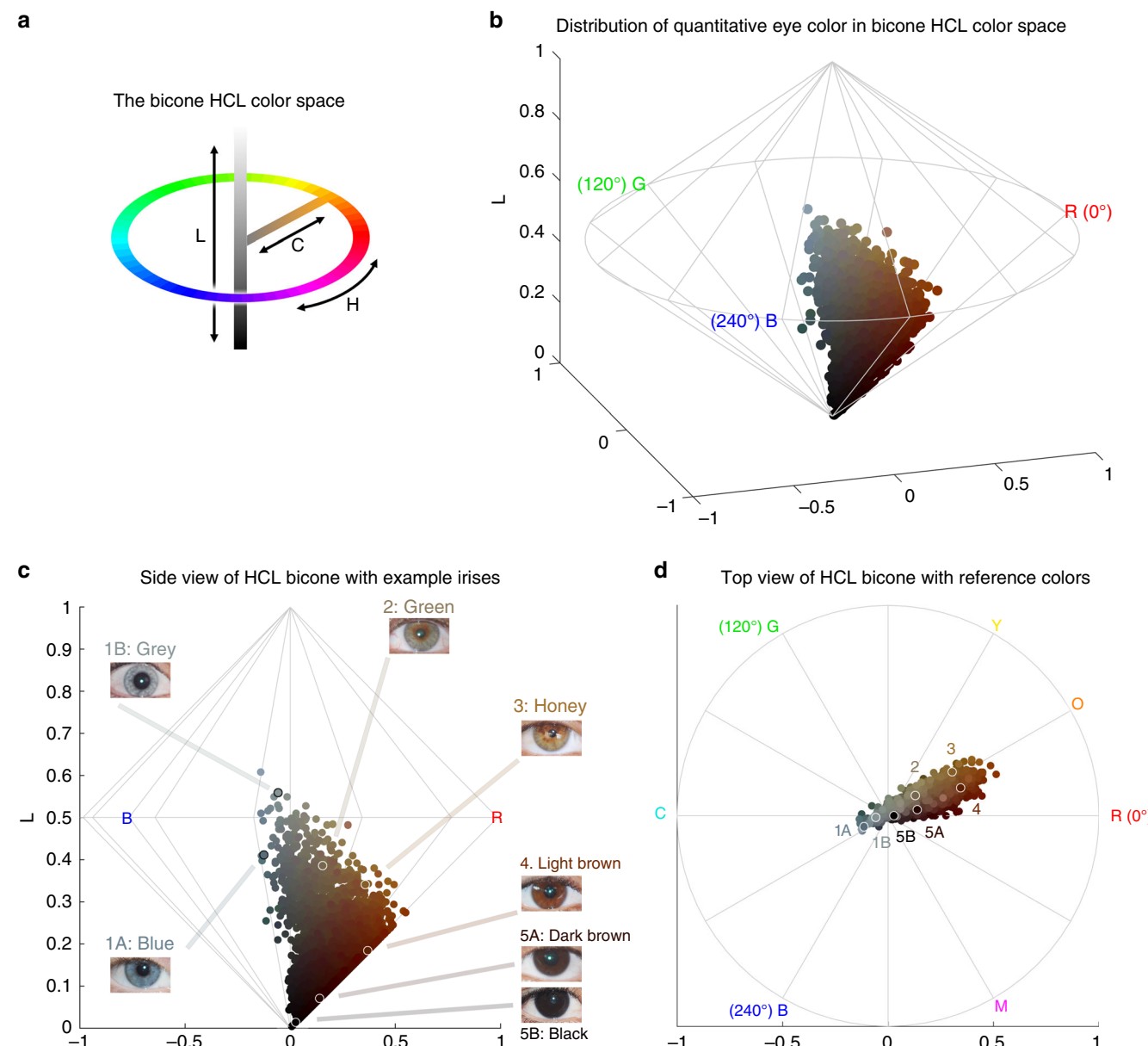

**Fig. 2** Quantitative assessment of eye pigmentation in the CANDELA sample. **a** Three-dimensional distribution of quantitatively assessed iris colors in the bicone HCL (hue, chroma, lightness) color space. Each dot corresponds to a CANDELA individual and its color represents the average iris color for that person. The color space has a polar coordinate system, where the vertical axis represents L (lightness/brightness, from dark = 0 to light = 1), the horizontal distance from the central axis represents C (chroma/saturation, from desaturated = 0 to fully saturated = 1), and H (hue/ tone) represents the angle when a vertical plane is rotated along the central axis (the three primary colors red (R), green (G) and blue (B) being situated at angles of 0°, 120° and 240° respectively). **b** The full range of the HCL color space, showing how the three color components vary in the space. Hue varies as a color circle, coming back to red at 360°. The unlabeled axes represent the Cartesian equivalents for the C and H variables, which define a polar coordinate system, as shown in panel **a**. **c** Side view of the bicone in **a** showing how the L (lightness/brightness) and C (chroma/saturation) of eye colors vary among CANDELA volunteers. The position of the dots corresponding to the average eye colors of the sample images in Fig. 1c are indicated. **d** Top view of the bicone in **a** showing how H varies among the eye colors of CANDELA volunteers. The position of the dots corresponding to the average color of the sample images in Fig. 1c are highlighted by white circles. In addition to the primary RGB colors, the secondary colors orange (O), yellow (Y), cyan (C) and magenta (M) are shown at their corresponding H angles

performed GWAS analyses conditioned on six well-established pigmentation SNPs, which explain a large proportion of the phenotypic variance seen in our sample (Supplementary Table 5 and Methods): rs16891982 (*SLC45A2*), rs12203592 (*IRF4*), rs10809826 (*TYRP1*), rs1800404 (*OCA2*), rs12913832 (*HERC2*) and rs1426654 (*SLC24A5*). The association statistics showed no evidence of residual population stratification, except for skin pigmentation (genomic inflation factor λ = 1.11) (Supplementary Table 4A and Supplementary Figure 6B). We interpret this as

resulting from a relatively high polygenicity of skin pigmentation, rather than from residual population stratification, as has been suggested by other studies[2,4,23,24]. Consistent with this view, an analysis based on the Tail Strength statistic[25] indicates modest but significant polygenicity for all the traits examined, with the highest values being observed for skin pigmentation (see Supplementary Table 4A and Methods).

Across all traits, we detected genome-wide significant association (P values < 5 × 10$^{-8}$) at SNPs in 12 genome regions (Table 1,

**Table 1 Features of index SNPs in genome regions associated with pigmentation traits in the CANDELA sample**

| | | | | Trait/Association (P value) | | | | | |
|---|---|---|---|---|---|---|---|---|---|
| | | | | Skin | Hair | Eye | | | |
| Region | SNP | Candidate gene | SNP annotation | MI | Categorical | Categorical | L (brightness) | C (saturation) | cos(H) (hue) |
| **1q32** | **rs3795556** | **DSTYK** | **3' UTR** | 2.1E-01 | 9.1E-01 | 6.8E-01 | 6.9E-03 | **4.0E-09** | 2.3E-01 |
| 5p13 | rs16891982[a,b] | SLC45A2 | F374L | **1.3E-117** | **6.3E-66** | **1.3E-15** | **4.0E-17** | **5.4E-07** | 1.8E-04 |
| 6p25 | rs12203592[b] | IRF4 | Intronic | **3.2E-10** | **2.0E-13** | **1.3E-12** | **3.2E-14** | 1.1E-03 | 4.5E-02 |
| 9p23 | rs10809826[a,b] | TYRP1 | Intergenic | 1.1E-03 | 3.3E-02 | **1.0E-10** | **5.0E-16** | **2.0E-08** | 1.2E-02 |
| **10q26** | **rs11198112** | **EMX2** | **Intergenic** | **1.7E-10** | 6.1E-01 | 3.6E-01 | 4.9E-01 | 7.7E-01 | 4.9E-01 |
| 11q14 | rs7118677[a,c] | GRM5 | Intronic | **1.1E-09** | **3.1E-06** | 6.1E-01 | 7.5E-01 | 4.8E-01 | 5.5E-01 |
| 11q14 | rs1042602 | TYR | S192Y | **9.1E-10** | **2.3E-06** | 7.5E-01 | 3.9E-01 | 3.6E-02 | 7.8E-01 |
| 11q14 | rs1126809[a,c] | TYR | R402Q | **2.5E-09** | **6.2E-06** | 1.2E-04 | **5.3E-06** | 7.4E-02 | 7.7E-04 |
| 15q13 | rs4778219[c] | OCA2 | Intronic | 8.3E-01 | 7.4E-01 | 4.7E-02 | 8.9E-02 | 6.2E-01 | 2.0E-01 |
| 15q13 | rs1800407[c] | OCA2 | R419Q | **6.5E-09** | 5.5E-02 | 1.1E-02 | 7.2E-02 | **1.4E-07** | **4.8E-06** |
| 15q13 | rs1800404[b] | OCA2 | Synonymous/TFB | **5.0E-11** | 1.3E-01 | **1.3E-11** | **5.0E-19** | **1.2E-06** | 4.1E-02 |
| 15q13 | rs12913832[b] | HERC2 | Intronic | **1.0E-17** | **7.9E-105** | **1.0E-200** | **1.0E-200** | **5.7E-07** | **1.3E-92** |
| 15q13 | rs4778249[a,c] | HERC2 | Intronic | **2.5E-06** | 1.2E-03 | **1.4E-10** | **2.5E-20** | **4.2E-15** | 5.1E-01 |
| 15q21 | rs1426654[b] | SLC24A5 | T111A | **1.6E-130** | **1.0E-18** | **1.0E-26** | **7.9E-50** | **6.3E-45** | 4.4E-01 |
| 16q24 | rs885479 | MC1R | R163Q | **1.9E-07** | 5.4E-02 | 5.6E-01 | 9.6E-01 | 8.0E-01 | 9.0E-01 |
| **19p13** | **rs2240751** | **MFSD12** | **Y182H** | **1.7E-10** | 8.2E-01 | 3.1E-01 | 9.6E-01 | 1.2E-01 | 9.1E-01 |
| **20q13** | **rs17422688** | **WFDC5** | **H97Y** | 5.2E-01 | 6.9E-01 | 8.2E-01 | 2.0E-01 | 9.0E-01 | **2.0E-08** |
| **22q12** | **rs5756492** | **MPST** | **Intronic** | 4.6E-03 | 9.9E-01 | 2.7E-02 | 9.5E-03 | **5.0E-08** | 1.5E-01 |

Novel genomic regions are in bold. Genome-wide significant P values (<5 × 10⁻⁸) are in bold and underlined. Genome-wide suggestive significant P values (<10⁻⁵) are in bold
MI: melanin index, L: lightness, C: chroma, H: hue
[a]These SNPs were obtained through imputation. Their imputation quality 'info' metric was ≥0.975, the median value being 0.993. The other SNPs were obtained from chip genotyping, and their 'concordance' metric was >0.9, the median value being 0.981
[b]These SNPs have been robustly associated with pigmentation traits in previous studies, and they explain a large proportion of the phenotypic variance in our sample (see Methods). These six SNPs were therefore used to condition the GWAS in subsequent analyses
[c]The independence of association signals of these SNPs from the main index SNPs in the same regions was confirmed by conditioned analyses

Fig. 3 and Supplementary Figure 7). As expected from the gain of power provided by conditioning on known pigmentation loci with large effects in our sample, P values from the conditioned analyses (Supplementary Table 5) are smaller for each loci than those obtained in the unconditioned analyses (Table 1). This includes well-established pigmentation SNPs not used in conditioning (rs1042602 in TYR, rs885479 in MC1R; Table 1, Supplementary Table 5), which are expected to represent confirmed associations (the association P value for rs885479 in MC1R with skin pigmentation was only suggestive in the unconditioned analyses but became genome-wide significant in the conditioned analyses). Furthermore, in the unconditional analysis the novel association in DSTYK was genome-wide significant only with eye color variable C, but in the conditional analysis this association is also genome-wide significant for eye color variable L.

Altogether, skin pigmentation showed association with SNPs in eight regions, of which: (i) five have been robustly replicated in previous studies in Europeans or East Asians[26–29]; (ii) one (19p13) has recently been associated with skin pigmentation in Africans[5], but at different SNPs than seen here; and (iii) one (10q26) has not been previously reported. SNPs at four of the skin pigmentation regions were also found to be significantly associated with eye and hair color (in 5p13, 6p25, 15q13 and 15q21; Table 1). In addition, eye color shows association with SNPs in four other regions (in 1q32, 9p23, 20q13 and 22q12), of which three (in 1q32, 20q13 and 22q12) have not previously been reported. The genomic regions associated with categorical eye color showed stronger association with the quantitative eye color variables (Table 1), consistent with the greater statistical power for association testing of the quantitative color variables extracted from the digital photographs, compared with the categorical variables.

Other than these primary genome-wide SNP association tests, we performed two types of secondary analyses. Firstly, we examined association for each index SNP in the newly associated regions (i.e., the variant with the lowest P value within a region) in each country sample separately, and combined results as a meta-analysis (Supplementary Figure 8). For all SNPs, significant effects were in the same direction in all country samples, the variability of effect reflecting sample size. Secondly, we combined all phenotypes in a single multivariate association analysis, seeking to exploit the correlation between traits (Supplementary Table 6). As expected, index SNPs with effects across phenotypes were found to be significantly associated in this combined analysis (P value < 5 × 10⁻⁸), whereas SNPs that only affected one trait were not associated at genome-wide significance, consistent with a reduced power under this scenario[30].

We evaluated the presence of multiple, independent, signals of association at each genomic region highlighted in the primary GWAS by performing step-wise regression (using the same model as in the primary analyses), conditioning on the index SNP at each region (Table 1). Evidence of genome-wide significant association was abolished for all regions except 11q14 and 15q13, where a total of three and five independent signals were detected, respectively (Table 1). These two regions include, respectively, the GRM5/TYR and OCA2/HERC2 genes. SNPs in these regions have been robustly associated with pigmentation traits by previous analyses, including a number of GWAS and candidate gene studies[4,27,31–52]. However, since the SNPs examined in those reports often differ, the independence of these SNPs' effects has not been systematically evaluated. Consistent with our findings, two independent signals of association in 11q14 have been reported in a GWAS for skin pigmentation in the African/European admixed population of Cabo Verde[32]. Seven of the eight independently associated SNPs detected here impact on skin pigmentation (the exception being rs4778249 in15q13). In addition to the effect on skin pigmentation of the three associated SNPs in GRM5/TYR, two (rs1042602 and rs7118677) were also associated with hair pigmentation, and one (rs1126809) with eye color (Table 1). The five independently associated SNPs in OCA2/HERC2 impact on eye color variation, with one of these SNPs also impacting on hair color (rs12913832). Genome

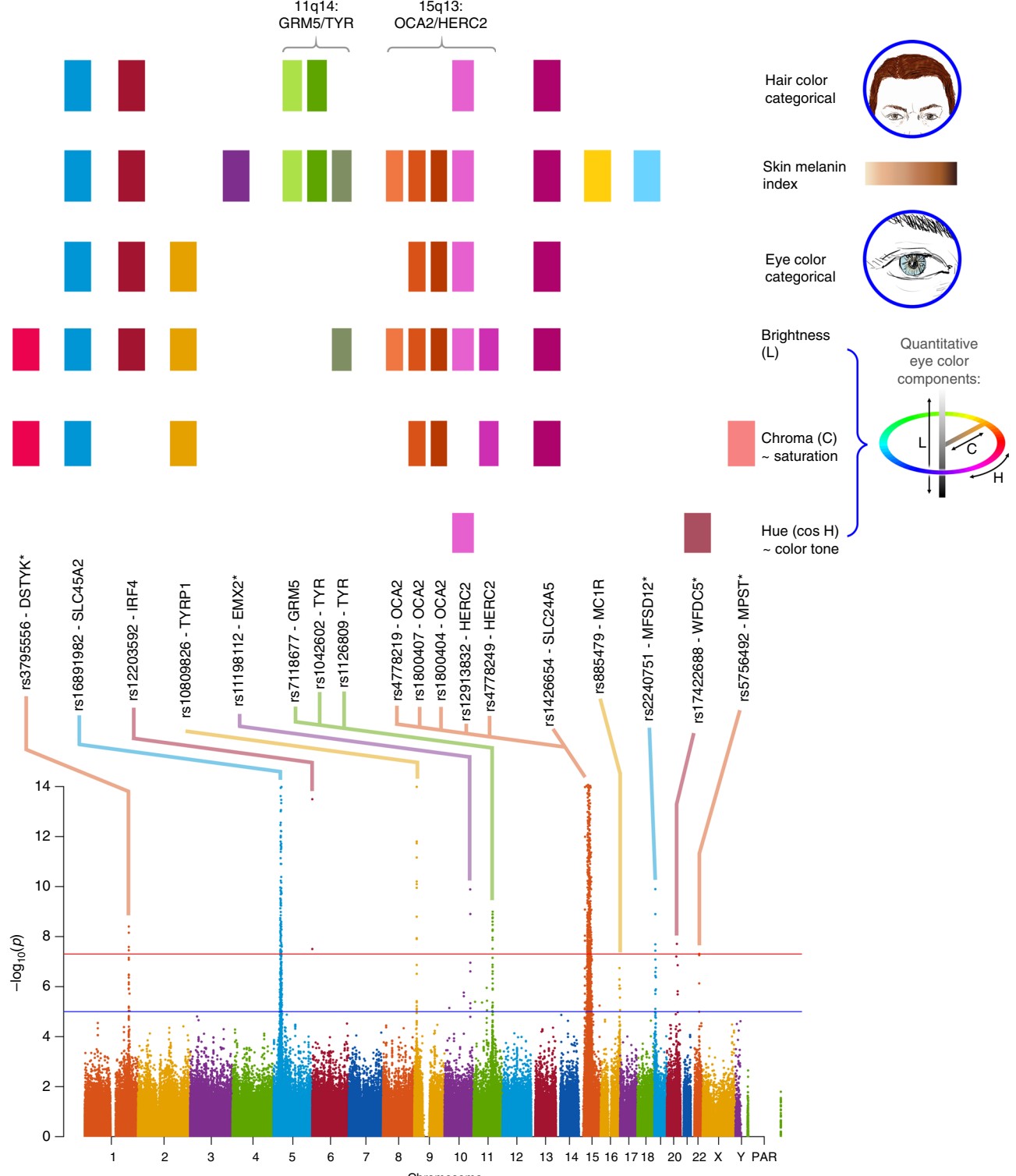

**Fig. 3** Summary of GWAS findings. Results are presented for six pigmentation traits: skin melanin index (MI, quantitative), categorical hair color, categorical eye color, and three quantitative eye color variables extracted from digital photographs: L (lightness/brightness), C (chroma/saturation) and cos H (cos hue/tone). These traits are represented on the right. The HCL color space with the three axes of variation is shown in the inset. To provide a global summary of the results, a composite Manhattan plot is presented at the bottom combining significant signals for all the traits. Horizontal lines indicate the suggestive (blue line, $P$ value $= 1 \times 10^{-5}$) and significant (red line, $P$ value $= 5 \times 10^{-8}$) thresholds. The y-axis was truncated at $-\log_{10}(P$ value$) = 14$. Index SNPs in each region are listed above the Manhattan plot. The association of these SNPs with specific traits is represented by colored boxes at the top: a box is shown if a SNP is associated with that trait (Table 1). Box colors correspond to colors assigned to each chromosome in the Manhattan plot, with slight variation when multiple independent hits were observed on the same chromosome. Novel genomic regions are marked with an asterisk

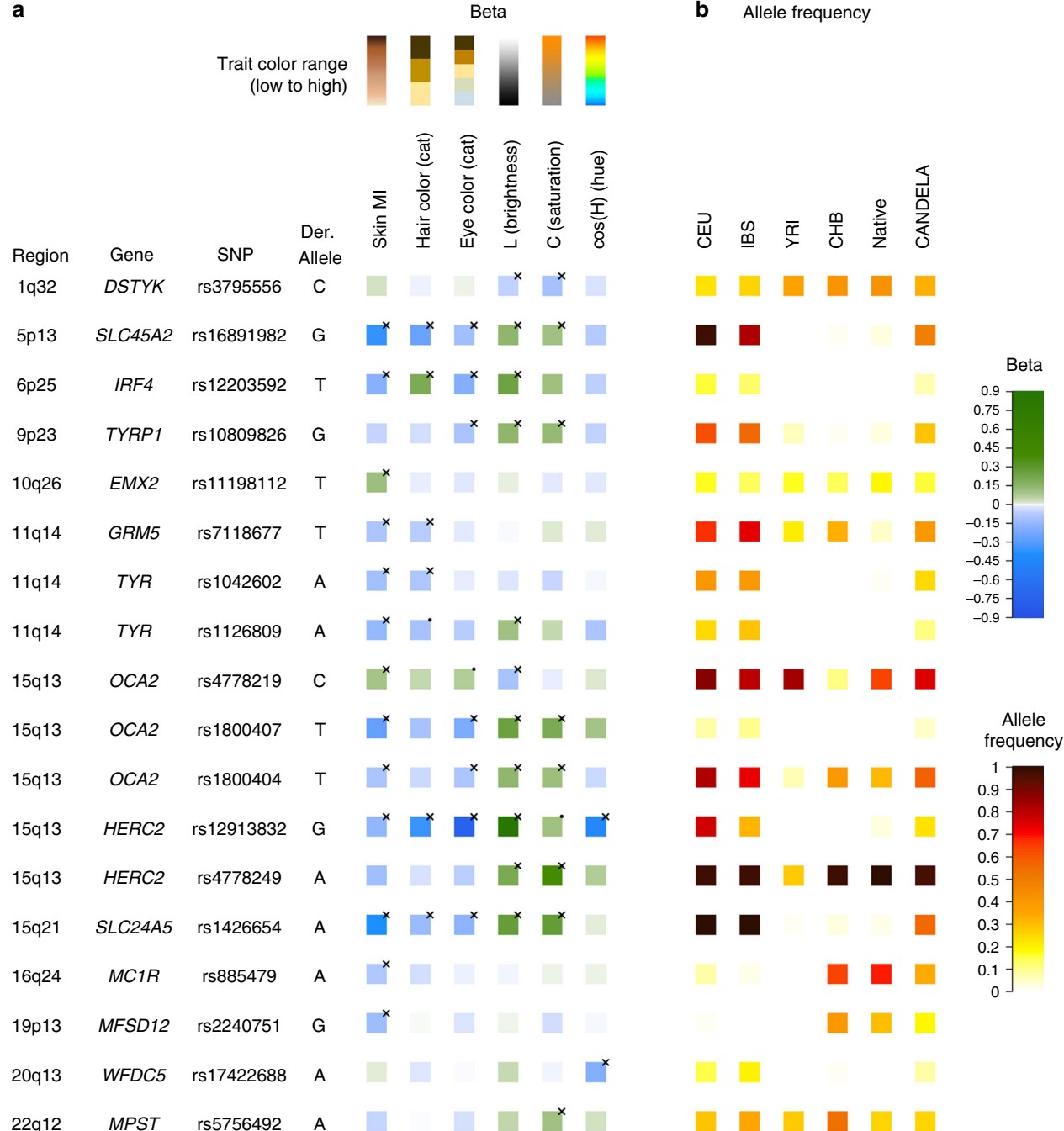

**Fig. 4** Phenotypic effects (regression beta-coefficients) and derived allele frequencies for the 18 index SNPs showing independent association in the CANDELA sample (Table 1). In **a** traits are shown at the top, with illustrative color ranges. Beta-coefficients have been standardized to facilitate comparison across traits. Positive betas are shown in green and negative betas in blue (with color intensity reflecting beta values as indicated on the scale to the right). Significant betas are marked with a cross. In **b** allele frequencies are shown for the CEU, IBS, CHB and YRI samples from the 1000 Genomes Project Phase 3, the CANDELA sample and Native Americans (from Reich et al.[17] and Chacon-Duque et al.[79]). On the right is shown the color scale used to represent allele frequencies (Supplementary Table 7)

annotations suggest that these eight independently associated SNPs could have separate functional relevance (Table 1). Four occur in exons, of which three result in non-conservative amino-acid substitutions and one (rs1800404) encodes a synonymous substitution (in exon 10 of *OCA2*) and is located in a conserved binding site for transcription factor YY1 (known to regulate pigmentation in animal models[33]). The allele associated with lighter skin pigmentation at rs1800404 has also been associated with a shorter *OCA2* gene transcript that is missing exon 10 and codes for a protein

missing a transmembrane region[5]. The other four independently associated SNPs are located in introns of *GRM5/TYR* or *OCA2/HERC2*. For one of these (rs12913832), intronic within *HERC2*, there is strong experimental evidence indicating that it regulates transcription of the neighboring *OCA2* gene[34].

Figure 4 summarizes the allelic effects and derived population allele frequencies for the 18 index SNPs identified here. Most of these show large differences between continental populations, with the frequency in the CANDELA sample being intermediate,

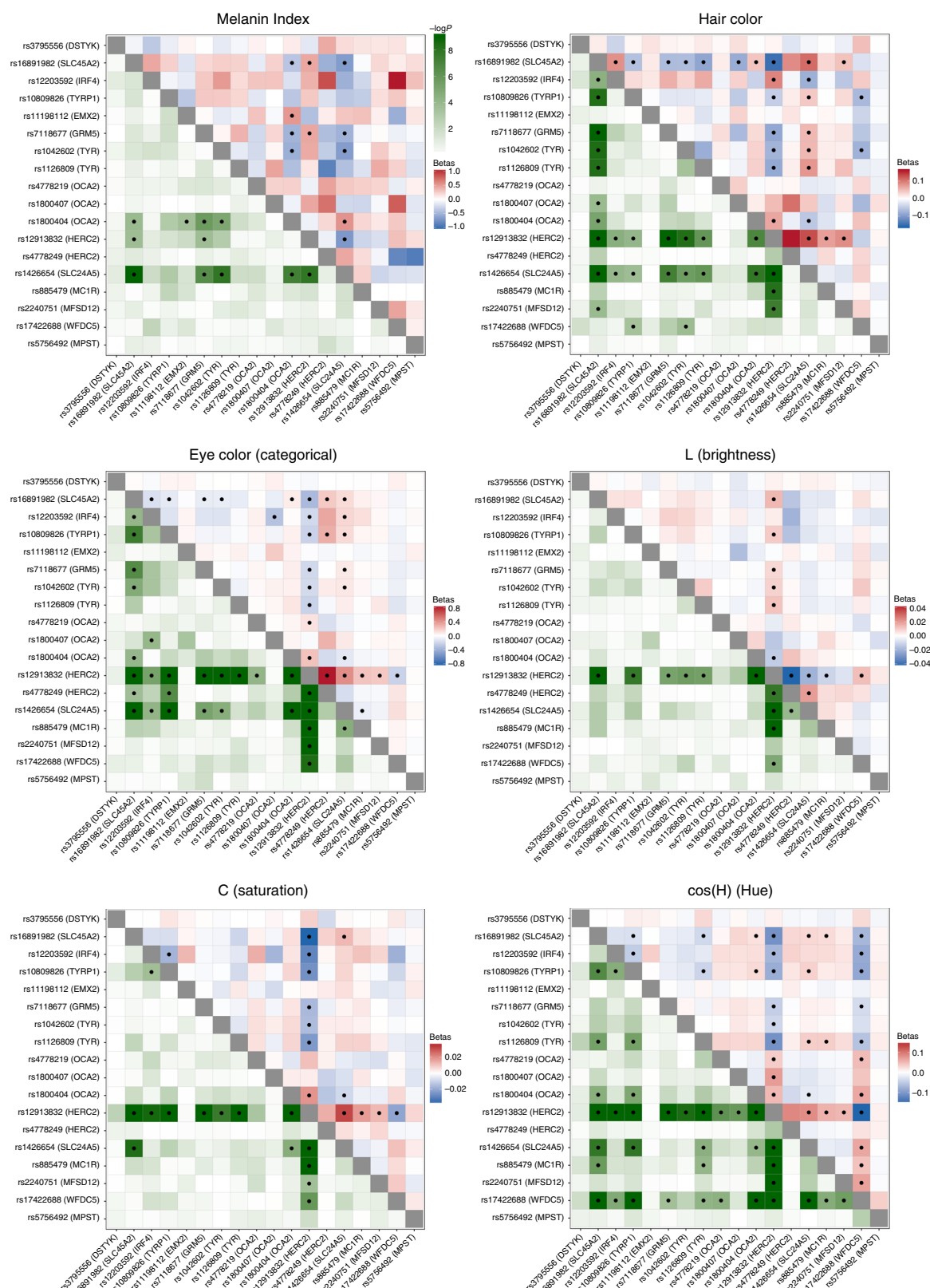

**Fig. 5** Heatmaps of statistical interactions between the 18 index SNPs identified here. Each panel corresponds to a different trait. The lower left triangle represents $-\log_{10} P$ values for the interaction term included in the regression model (with the color scale shown at the top). The upper right triangle represents regression beta-coefficients for each interaction term, colored from blue (negative effect) to white (no effect) to red (positive effect). As the scale for each trait is different, separate scales for effect sizes are shown next to each panel. Interactions that are significant (after Bonferroni correction) are marked with a black dot

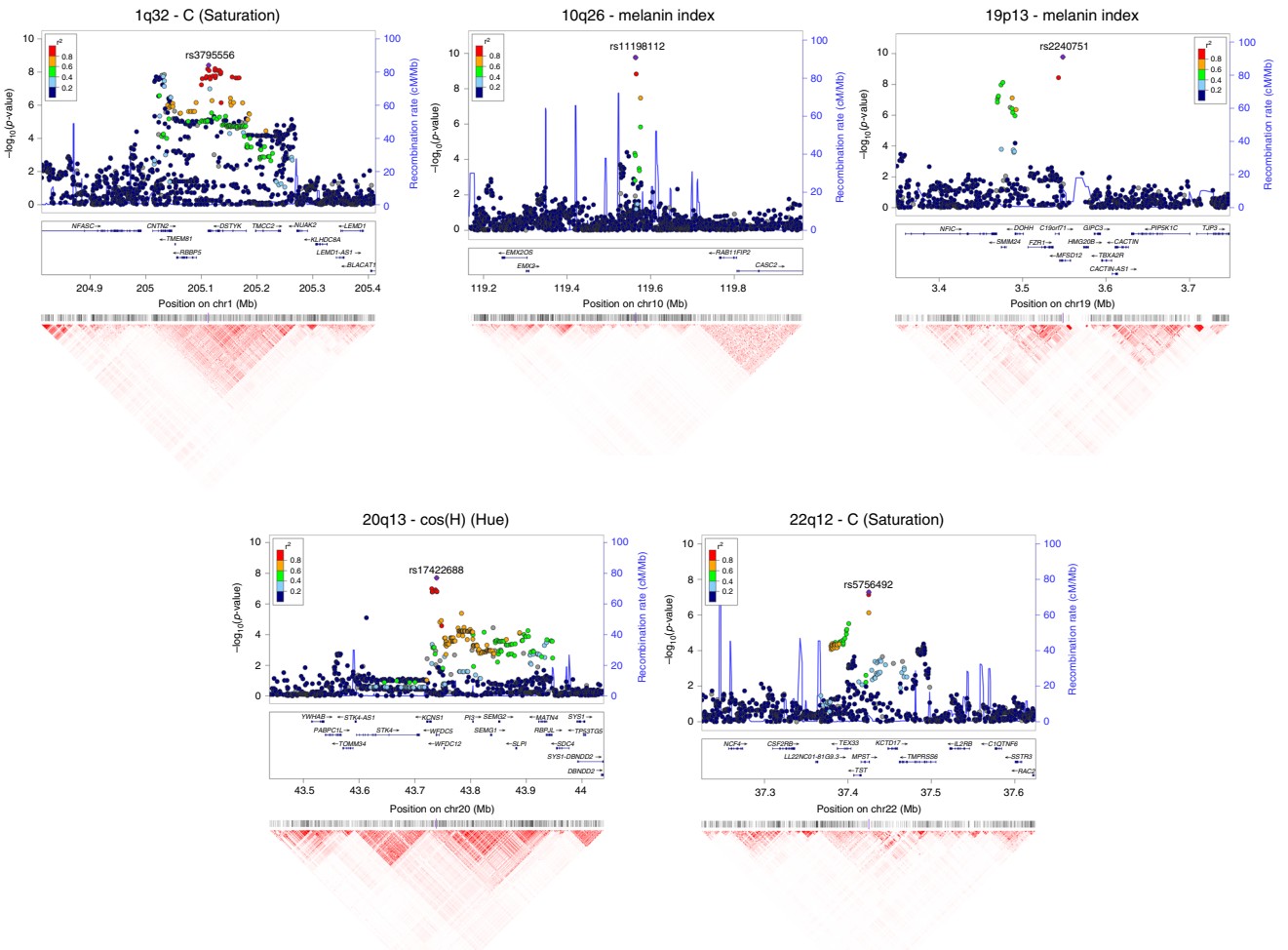

**Fig. 6** Regional association (LocusZoom) plots for SNPs in the five genomic regions showing novel genome-wide significant associations to pigmentation traits. Chromosomal location and trait are specified in the title of each panel. In each region, index SNPs (Table 1) are highlighted with a purple diamond. Colors for other SNPs represent the strength of LD between that SNP and the index SNP (in the 1000 Genomes AMR data). Local recombination rate in the AMR data is shown as a continuous blue line (scale on the right y-axis). Genes in each region, their intron–exon structure, direction of transcription and genomic coordinates (in Mb, using the NCBI human genome sequence, Build 37, as reference) are shown in the middle of each panel. At the bottom is shown a pairwise LD heatmap across all SNPs in a region (using $r^2$, ranging from red indicating $r^2 = 1$ to white indicating $r^2 = 0$)

consistent with its admixed ancestry. For all but three SNPs (rs3795556, rs11198112 and rs4778219), the derived allele is associated with lower pigmentation.

**Interaction of SNPs independently associated with pigmentation**. We examined interaction between the index SNPs of Table 1 by testing regression models including all possible pairs of SNPs, adjusting for age, sex and the first six PCs, as in our primary association analysis. A number of significant interactions were detected at a multiple-testing corrected P value threshold of $3.3 \times 10^{-4}$ (Fig. 5). A different pattern of interactions was observed for skin, relative to hair or eye pigmentation. In the case of skin pigmentation, significant interactions were seen mainly between SNPs that, individually, have strong effects (in *SLC45A2*, *SLC24A5*, *HERC2/OCA2* and *TYR/GRM5*). By contrast, for hair and eye color, SNPs in the regions with strongest individual effects (*SLC45A2*, *SLC24A5* and *HERC2/OCA2*) showed significant interaction with SNPs at most other pigmentation-associated regions. This included regions that individually do not have a significant effect on a particular trait (e.g., *MC1R* and *MFSD12* with hair or eye pigmentation, respectively). These

results are in line with other analyses of epistasis for pigmentation traits[35,36].

**Candidate genes in genome regions showing novel association signals**. The 10q26 region that is newly associated with skin pigmentation shows genome-wide significant association with a linkage disequilibrium (LD) block of SNPs spanning ~100 Kb, within an intergenic region of ~400 Kb (Fig. 6). Genome annotations indicate that this region overlaps an open chromatin segment that is highly conserved evolutionarily and includes several transcription factor binding sites (Supplementary Figure 9). The derived allele for the index SNP (rs11198112) is associated with darker skin pigmentation, in contrast to the effect of the majority of variants associated with skin pigmentation (Fig. 4). The derived allele is segregating at low to moderate frequency across many populations, but reaches its highest frequency (>50%) in Native Amazonians and Melanesians (Supplementary Figure 10). The index SNP is included in the binding site for transcription factor EBF1 (early B-cell factor). If the effect of this SNP is mediated through regulation of nearby genes, of potential interest is the gene encoding for the *EMX2* transcription factor (*empty spiracles homeobox 2*), which flanks the associated

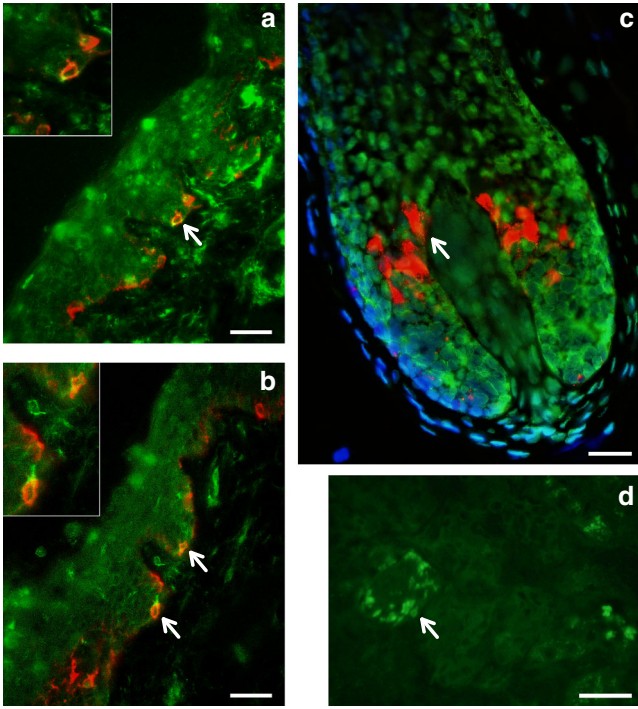

**Fig. 7** Immunohistochemical analysis of *MFSD12* protein expression in the epidermis of human scalp. MFSD12 expression (green fluorescence) was detected in multiple skin cell types (**a**, **b**). MFSD12 expression levels were higher in melanocytes (identified with an anti-melanocyte antibody in red fluorescence) than in adjacent keratinocytes (green only). Co-localization of both *MFSD12* and the melanocyte-specific protein gp100 expression can be seen in yellow/orange fluorescence (arrow). Insets show higher magnification views of arrowed *MFSD12*-expressing melanocytes in skin epidermis. **c** A proportion of keratinocytes in scalp hair follicle from the same tissue also expressed *MFSD12*(green only). By contrast with the skin, *MFSD12* expression was not detected in hair melanocytes (i.e., seen as red fluorescence only indicating gp100 protein expression). **d** Positive control (human kidney). Note *MFSD12* expression in kidney tubular cells (arrow). Scale bars: **a**, **b** = 50 μM. **c** = 15 μM, **d** = 30 μM

region (Fig. 6). Mouse experiments have shown that Emx2 regulates the expression of *Mitf* (a key regulator of melanocyte development and survival) as well as of *Tyr* and *Tyrp-1* (two melanocyte-specific genes responsible for melanin production)[37]. In addition, this gene has been recently associated to tanning response in Europeans[38].

SNPs showing genome-wide significant association in the 19p13 region span ~100 Kb and show strongest association for SNP rs2240751 located in the third exon of the *major facilitator superfamily domain containing 12* (*MFSD12*) gene (Table 1, Fig. 6). Variants in this region have recently been associated with skin pigmentation in Sub-Saharan Africans[5]. The index SNP in the CANDELA data (rs2240751) leads to a tyrosine for histidine substitution at amino-acid 182 of *MFSD12* (Y182H), which is common in East Asians and Native Americans but rare elsewhere (Fig. 4, Supplementary Table 7, Supplementary Figure 11). This variant occurs in a highly conserved sequence (as indicated by Genomic Evolutionary Rate Profiling (GERP) and Site-specific Phylogenetic (SiPhy) metrics) and the replacement of a polar for a basic amino acid could affect the function of the protein, as indicated by low Sorting Intolerant from Tolerant (SIFT; <0.01) and high Polymorphism Phenotyping v2 (PolyPhen2; >0.99) scores. Functional analyses indicate that *MFSD12* is involved in lysosomal biology and that it can alter pigmentation coloration in animal models[5]. Since *MFSD12* is highly expressed in melanocytes

relative to other cell types[5], and is also expressed in human skin (Supplementary Figure 12C), we examined the cellular expression of *MFSD12* in normal human skin using immunohistochemistry. *MFSD12* was detected in the cytoplasm of a subpopulation of melanocytes in the epidermis (Fig. 7), possibly reflecting expression of this protein at a particular maturation stage of skin melanocytes. By contrast, no expression was detected in hair bulb melanocytes of anagen scalp hair follicles.

Of the three novel regions associated with quantitative digital eye color variables, the one in 1q32 is characterized by substantial LD over a region of ~300 Kb (Fig. 6) and is associated with the L and C variables (Table 1 and Fig. 4). Strongest association is seen for markers overlapping the *DSTYK* gene (*dual serine/threonine and tyrosine protein kinase*), the index SNP (rs3795556) being located in the 3' untranslated region of the *DSTYK* transcript. Expression studies have shown that *MITF* regulates the expression of *DSTYK* in melanocytes[39]. The 20q13 region associated with the cos(H) variable shows strong LD over a region of ~200 Kb. Strongest association is seen for SNPs overlapping the *WFDC5* gene (WAP Four-Disulfide Core Domain 5, Fig. 6), with the index SNP (rs17422688) leading to a histidine for tyrosine substitution (H97Y) in a highly conserved region (based on GERP and SiPhy conservation metrics). This amino-acid change is predicted to affect protein function, as implied by low SIFT (0.03) and high PolyPhen2 (0.81) scores. *WFDC5* is highly expressed in skin tissues (Supplementary Figure 12D). Several WAP Four-Disulfide Core Domain genes have been shown to be expressed in the human iris[40]. SNPs in 22q12 associated with the C variable show LD over a region of ~100 Kb (Fig. 6). The index SNP (rs5756492) is located in the second intron of the gene encoding *Mercaptopyruvate sulfurtransferase* (*MPST*), an enzyme playing a role in cyanide detoxification[41] and cellular redox regulation[42]. *MPST* is expressed in the skin (Supplementary Figure 12E) and the human iris[40].

**Evidence for selection at pigmentation-associated regions**. Previous studies have detected signatures of selection around several pigmentation genes[10,11,43]. In agreement with those analyses, we found strong signals of selection in Europeans (CEU) and East Asians (CHB) from the 1000 Genomes (1KG) Project at most of the pigmentation-associated regions replicated here (Supplementary Figure 13 and Supplementary Table 8). Often the associated SNPs do not show the strongest selection signals, which suggests that selection may have acted on other nearby SNPs (Supplementary Figure 13). Highly significant signals of selection were also detected in three of the five novel pigmentation regions identified here, with the strongest signals being observed in the *MFSD12* region in East Asians (Fig. 8a). More generally, we also detected a significant enrichment of maximum Population Branch Statistic (PBS) and Integrated Haplotype Score (iHS) scores at genomic regions showing at least suggestive association (i.e., those including SNPs with P values < $10^{-5}$) compared to the rest of the genome (Supplementary Table 9).

Selection for skin pigmentation has been proposed to relate to adaptation to solar radiation[8]. Consistently, a correlation between allele frequencies at certain skin pigmentation-associated SNPs with solar radiation levels has been reported in the Human Genome Diversity Project (HGDP) population panel[44,45]. We re-evaluated this correlation for the index SNPs of Table 1 in a dataset we compiled including 64 native populations from around the world (excluding the HGDP panel; Supplementary Table 10). Allele frequencies at four SNPs showed a significant correlation with solar radiation (Supplementary Table 11). Three of these SNPs are in gene regions replicated in the CANDELA sample (rs12913832 and rs1800404 in the *HERC2/OCA2* gene region and rs885479 in *MC1R*). The fourth is the index SNP at *MFSD12*

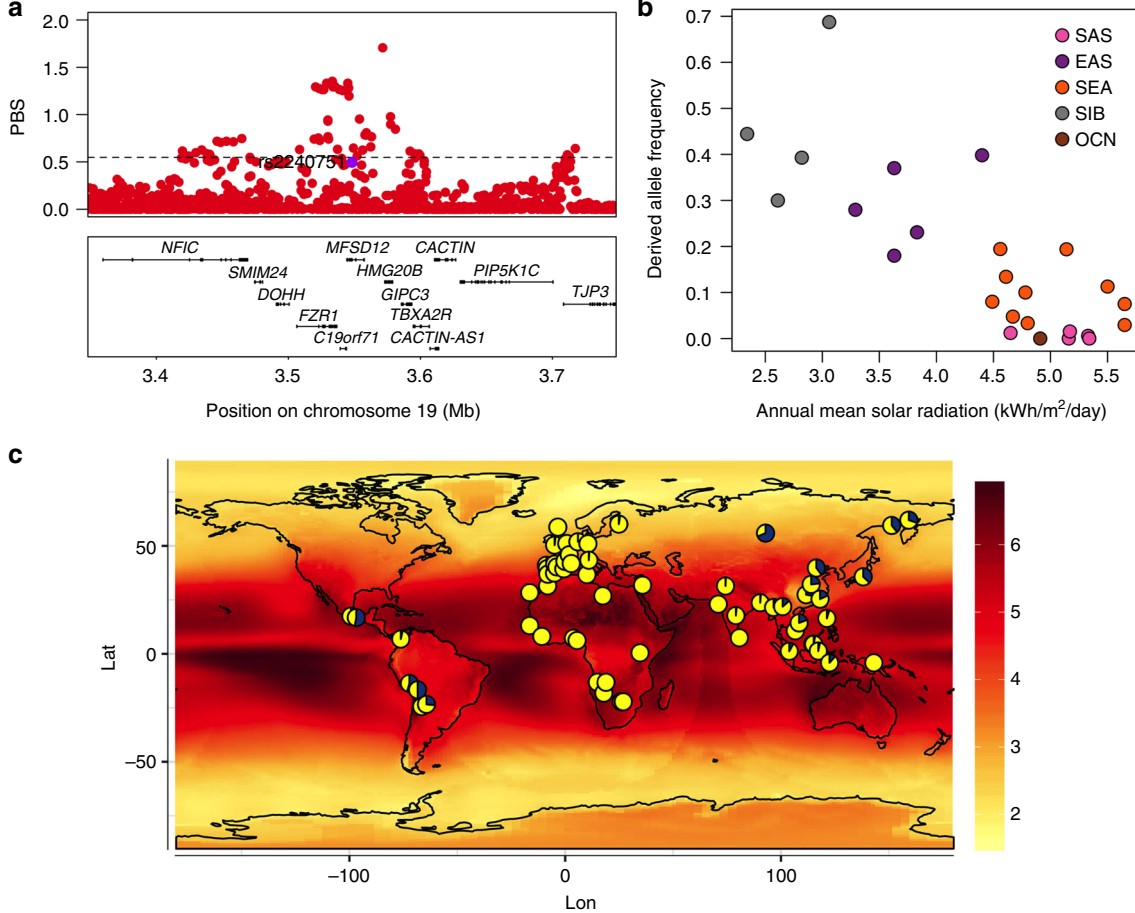

**Fig. 8** Evidence for selection in the MFSD12 gene region. **a** PBS scores in the 1000 Genomes CHB sample for SNPs across the region (index SNP rs2240751 is highlighted in purple and the horizontal black line represents the 99th percentile threshold). **b** Plot of the derived allele frequency at rs2240751 against mean annual solar radiation in Easter Eurasian populations. Populations are abbreviated as follows: SAS South Asians, EAS East Asians, SEA South East Asians, SIB Siberians, OCN Oceanians. **c** Allele frequencies at rs2240751 in 64 native populations from across the world mapped onto solar radiation. Pies charts are centered at the approximate geographic location of each population with the derived allele frequency represented in blue. Geographic coordinates, sample size, mean annual solar radiation and the frequency of the derived allele for each population are shown in Supplementary Table 10 and Supplementary Figure 11

(rs2240751), which showed a strong correlation with solar radiation in Eastern Eurasia ($\log_{10}(BF) = 2.32$, $P$ value = 0.004; $\rho = -0.28$, $P$ value = 0.047) (Fig. 8b, c).

Considering the evidence for selection in the *MFSD12* region in Eastern Eurasians, we estimated the time since the start of selection and the selection coefficient for this region in the CHB dataset from 1KG using an approximate Bayesian computation (ABC) approach (Supplementary Figure 14, 15 and 16 and Methods). We obtained a median estimate for the selection coefficient of 1.15% (95% credible interval 0.08%–4.4%) and a median age for the start of selection of 10,834 year ago (95% credible interval of 5266–33,801 years ago).

## Discussion

The analyses presented here highlight the complex genetic architecture of pigmentation variation in Latin Americans, with multiple gene regions as well as multiple independent variants at the *OCA2/HERC2* and *GRM5/TYR* regions, and several epistatic interactions, affecting pigmentation variation. Since the history of the New World involved the extensive admixture of Native Americans, Europeans and Africans, it is to be expected that variants impacting on pigmentation in those continental populations are segregating in Latin America. Further, since Native Americans trace their ancestry to East Asia, it is likely that certain

pigmentation variants present in Latin Americans should be shared with East Asians. Consistent with this scenario, we replicate 7 allelic variants that have been previously associated with pigmentation phenotypes in Europeans and one variant previously reported in East Asians (rs885479 in *MC1R*). It seems likely that we did not detect some of the other variants previously associated with pigmentation variation in Old World populations due to a combination of factors affecting power across studies. For instance, some of the reported variants could have high frequency in Old World populations that did not contribute to admixture in Latin America. Dissimilarities in phenotype assessment approaches and in trait definitions are also likely to explain some of the differences in association results across studies. For example, GWAS carried out in Europeans have mostly focused on variation in the brown to blue color spectrum. By contrast, the C (saturation) color component examined here, with which two new loci have been associated, captures variation within brown eyes (Fig. 2b) and the index SNPs at these loci have highest derived allele frequencies in East Asians (Fig. 4).

The convergent evolution of lighter skin pigmentation in Western and Eastern Eurasia stems partly from allelic heterogeneity at two well-established pigmentation genes initially identified in Europeans: *OCA2* and *MC1R*. In addition to allelic heterogeneity at these two genes, here we identify rs224071 at

*MFSD12* as another pigmentation variant specific to populations of East Asian/Native American ancestry. This gene region has been recently implicated in a study of skin pigmentation variation in Sub-Saharan Africans[5]. Strongest association in that study was seen for synonymous and intronic SNPs in *MFSD12*, variable only in Africans, and in an upstream regulatory region, variable in Africans and South and South East Asians, but not in Europeans or East Asians. By contrast, we found that in our sample the strongest association with skin pigmentation is seen for the Y182H amino-acid substitution in *MFSD12*, a variant seen at high frequency only in East Asians and Native Americans. It is thus likely that this variant rose in frequency in East Asia and was carried into the Americas during Native American migrations. This establishes *MFSD12* as an additional gene involved in the convergent evolution of lighter skin pigmentation in Eurasians. Furthermore, consistent with what is observed at several other pigmentation gene regions, we observe a strong signal of selection in this gene region in East Asians (dated after their split from Europeans), and a correlation of the frequency of the *MFSD12* Y182H variant with solar radiation levels in East Asia (Fig. 8). The pattern of variation at *MFSD12* is thus reminiscent of what is observed for certain pigmentation genes in Europeans (e.g., *OCA2* or *SCL45A2*). Associated SNPs at those genes are polymorphic mainly in Europeans, show strong signals of selection and a correlation of derived allele frequencies with latitude[44,45].

Our estimate of the selection coefficient for *MFSD12* is best viewed in the context of estimates for other pigmentation loci. Beleza et al.[32] used forward Monte Carlo simulations coupled with a rejection algorithm to estimate the selection coefficient at four pigmentation genes. Under an additive model, the selection coefficient for *KITLG* (rs642742 G allele) in Europe and East Asia was estimated to be 0.02, whereas the coefficients for *TYRP1* (rs2733831 G allele), *SLC45A2* (rs16891982 G allele) and *SLC24A5* (rs1426654 A allele) were estimated to be 0.03, 0.04 and 0.08, respectively. López et al.[46] estimated the selection coefficient of *SLC45A2* (rs16891982 G allele) to be 0.01 to 0.02 in a South European population. Similarly, using an ancient DNA forward simulation approach restricted to European populations, Wilde et al.[47] estimated the selection coefficient of *SLC45A2* (rs16891982 G allele), *TYR* (rs1042602 A allele) and *HERC2* (rs12913832 G allele) to be 0.03, 0.03 and 0.04, respectively. The selection coefficient that we estimated (0.01) for *MFSD12* thus lies at the lower end of those estimated for other pigmentation genes that appear to have been under selection. This result is in line with the relatively weaker phenotypic effect of *MFSD12*, relative to genes such as *SLC45A2* and *SLC24A5*. Our estimate for the age since the start of selection (10,833 ya (95% CI of 5266–33,801 ya)) suggests that it would have started long after the split of proto-East Asians from proto-Europeans.

Considering the evidence for solar radiation having contributed to shape the diversity of certain genomic regions in Old World populations, it is interesting that we do not detect pigmentation variants private to the Americas in the CANDELA sample (with the caveat that we might be unable to detect the effects of rare local variants). The American continent shows extensive variation in solar radiation levels as its territory extends along a North–South axis encompassing circumpolar and Equatorial latitudes (Fig. 8c). However, Native Americans do not exhibit a variation in skin pigmentation like that seen in Old World populations living at similar latitudes[48]. It has been suggested that this difference between continents might relate to cultural adaptations, environmental factors, or to another mechanism of biological adaptation, such as a better tanning ability[8,48]. It is possible that the lack of novel genetic adaptations to solar radiation levels in the Americas could relate to the relatively recent settlement of the New World, which started about 15,000 years ago. This recent settlement limits the time-span over which new genetic variants could have arisen and changed in frequency in response to selection pressures, particularly considering the magnitude of the selection coefficients that have been estimated for pigmentation associated loci.

## Methods

**Study subjects**. We analyzed data for 6357 individuals from the CANDELA sample, recruited in Brazil, Chile, Colombia, Mexico and Peru (Supplementary Table 1, http://www.ucl.ac.uk/silva/candela[16]). All volunteers provided written informed consent. Ethics approval was obtained from: Universidad Nacional Autónoma de México (México), Universidad de Antioquia (Colombia), Universidad Perúana Cayetano Heredia (Perú), Universidad de Tarapacá (Chile), Universidade Federal do Rio Grande do Sul (Brazil) and University College London (UK).

**Phenotype data**. A physical examination of each volunteer was carried out using the same protocol and instruments at all recruitment sites. Eye color was recorded in five categories (1-blue/gray, 2-honey, 3-green, 4-light brown, 5-dark brown/black). Hair color was recorded in four categories (1-red/reddish, 2-blond, 3-dark blond/light brown or 4-brown/black), as described in ref. [18]. Individuals with red hair were excluded prior to the analyses, as it is a rare in our sample (frequency of 0.6%) and this phenotype is known to stem from rare variants in *MC1R*. A quantitative measure of constitutive skin pigmentation (the MI) was obtained using the DermaSpectrometer DSMEII reflectometer (Cortex Technology, Hadsund, Denmark). The MI was recorded from both inner arms and the mean of the two readings used in the analyses. Measurements across the two arms were compared for each individual to assess variability of the MI measurement. The absolute difference between the two measurements was taken as the variability for an individual, and the median variability across all individuals was 1.03 units (Supplementary Figure 17). For comparison, the range of variation of MI in the CANDELA dataset is 20 to 65 units (in the QC-d set of individuals used for GWAS analyses). For visually inspecting the skin color distribution corresponding to variation in MI (Fig. 1a), MI values were converted to approximate RGB (red, green, blue) values (Supplementary Figure 18).

In addition to a direct assessment of eye color into four categories, we obtained quantitative variables related to eye color from digital photographs of the study subjects (taken following a standardized protocol as described in ref. [18]). One of the two eyes was selected based on image quality. Photographs were landmarked manually via a graphical interface designed in MATLAB (Supplementary Figure 1). Ten landmarks were used to delimit and extract the visible part of the iris. Additional landmarks were placed to select the whitest part of the sclera. This white reference and the darkest part of the pupil were used to normalize the image, adjusting for variable color casts or illumination levels across images. An adaptive threshold was then used to remove highlights such as reflections on the iris. The resulting images were individually checked for the presence of errors during the digitization steps leading to their exclusion. In total, 5513 iris images were retained for extracting RGB pixel color values.

A set of 195 photos were landmarked independently by two raters to assess inter-rater variability in extracted iris color. The median absolute difference between the RGB color values of the two raters across the whole set was 3.3 units (on a scale of 0–255).

The multivariate median of the RGB values across all pixels was calculated in order to obtain average RGB values for an iris (Fig. 2d, Supplementary Figure 2). Such RGB values, or their principal components (Supplementary Figure 3C), have been used in certain genetic association studies[49]. However, although the RGB color space is convenient for digital imaging, it is not necessarily the most appropriate in terms of human perception or biological relevance. Several other color spaces have therefore been considered in genetic studies of pigmentation. In particular, the HCL and CIE Lab color spaces have the advantage over RGB of being perception based[23,50]. Furthermore, it has been shown that melanosome density and the skin MI are strongly correlated with brightness (L)[51]. The main difference between the HCL and CIE Lab color spaces is that HCL, being directly derived from RGB, represents the three primary colors (red, green, blue) in opposing corners, while the CIE Lab represents four colors in different corners (red against green and blue against yellow). Since the HCL values in the CANDELA dataset occupy mainly the opposing red-orange and cyan-blue color hues (Fig. 2d), for this study we considered the HCL color space more informative than the nearly equivalent CIE Lab color space.

H is a circular variable representing color hue (tone) ranging from 0° to 360°, with red at 0°, green at 120° and blue at 240°. C (chroma or saturation) ranges from 0 (no color) to 1 (fully saturated color). L (lightness or brightness) ranges from 0 (black) to 1 (white). It was observed from the bicone color model (Fig. 2d) that the set HCL values lie approximately on a two-dimensional plane passing through the vertical central axis at an angle of ~20° (obtained from the circular median of H). H values were therefore standardized by subtracting 20°. Furthermore, since H is a circular variable, it was converted to cos(H) prior to its use for the analyses performed here. Cos(H) ranged from −1 (blue/gray eyes) to +1 (olive/brown/dark brown eyes). As the distribution of HCL values was nearly planar, sin(H) showed

comparatively little variation (equivalent to taking a projection onto the plane) and was ignored.

**Genotype data**. DNA samples from participants were genotyped on the Illumina HumanOmniExpress chip, which includes 730,525 SNPs. PLINK v1.9[22] was used to exclude SNPs and individuals with more than 5% missing data, markers with minor allele frequency <1%, related individuals with Identity-By Descent estimate (IBD) >0.1 (i.e., removing third-degree relatives (who have IBD 0.125) and higher) and those who failed the X-chromosome sex concordance check (sex estimated from X-chromosome heterozygosity not matching recorded sex information). After applying these filters, 669,462 SNPs and 6357 individuals were retained for further analysis. Due to the Native American, European and African admixture of the study sample (Supplementary Figure 5), there is inflation in Hardy–Weinberg $P$ values. We therefore did not exclude markers based on Hardy–Weinberg deviation, but performed stringent quality controls at software and biological levels (see also Supplementary Figure 14 from Adhikari et al.[52]). The SNP quality metrics generated from the GenCall algorithm in GenomeStudio were used for quality control. SNPs with low GenTrain score (<0.7), low Cluster Separation score (<0.3) or high heterozygosity values ((het. excess) > 0.5) were excluded[53]. The heterozygosity excess filter performs a function similar to a Hardy–Weinberg equilibrium check, but is more direct since it is based on the heterozygosity value, which unlike the $P$ value does not depend on sample size. Only SNPs that satisfy these criteria across all genotyping plates were retained[53]. The imputation 'concordance' score, which is a measure of poor genotyping quality, was also used to exclude some genotyped SNPs (see below). Finally, subsequent to the GWAS analyses (see below), the genotyping cluster plots for the index SNP identified were checked manually to verify genotyping quality.

**Genotype imputation**. The chip genotype data were phased using SHAPEIT2[54]. IMPUTE2[55] was then used to impute genotypes of untyped SNPs using variant positions from the 1000 Genomes Phase 3 data. The 1000 Genomes reference dataset included haplotype information for 1092 individuals across the world for 36,820,992 variant positions. Positions that are monomorphic in 1000 Genomes Latin American samples (Colombia, Mexico and Puerto Rico) were excluded, leading to 11,025,002 SNPs being imputed in our dataset. Of these, 48,695 had imputation quality scores <0.4 and were excluded. Median 'info' score (imputation certainty score) provided by IMPUTE2 for the remaining imputed SNPs was 0.986. The IMPUTE2 genotype probabilities at each locus were converted into most probable genotypes using PLINK v1.9[22] (at the default setting of <0.1 uncertainty). Imputed SNPs with >5% uncalled genotypes or minor allele frequency <1% were excluded. IMPUTE2 provides a 'concordance' metric for chip genotyped SNPs, obtained by masking the SNP genotypes and imputing it using nearby chip SNPs. Genotyped SNPs having a low concordance value (<0.7) or a large gap between info and concordance values (info_type0 – concord_type0 >0.1), two suggested indicators of poor genotyping quality, were also removed. Median concordance values of the remaining chip SNPs was 0.994. After quality control (QC), the final imputed dataset contained genotypes for 9,143,600 SNPs.

**Statistical genetic analyses**. Narrow-sense heritability (defined as the additive phenotypic variance explained by a genetic relatedness matrix (GRM) computed from the SNP data) was estimated using the software GCTA[56] by fitting an additive linear model with a random effect term whose variance is given by the GRM (with age and sex as covariates). The GRM was calculated using the LDAK software[19], which accounts for LD between SNPs. An LD-pruned set of 160,858 autosomal SNPs was used to estimate continental ancestry using the ADMIXTURE program[57] (Supplementary Figure 5). The correlation between traits and covariates was examined calculating Pearson's correlation coefficients (using R).

PLINK 1.9[22] was used to perform the primary association tests on the best-guess imputed genotypes (genotypes with the highest probability, i.e., the most probable genotypes) for each pigmentation phenotype using multiple linear regression. We used an additive genetic model incorporating age, sex and 6 genetic PCs as covariates. PCs were obtained from an LD-pruned dataset of 160,858 SNPs. Individual outliers (including individuals with >20% African or >5% East Asian ancestry, as estimated by ADMIXTURE) were removed and PCs recalculated after the removal of these individuals. The number of PCs to be included in the regression was determined by inspecting the proportion of variance explained and by checking scree and PC scatter plots (Supplementary Figure 6A).

Pigmentation is one of the best-characterized complex human traits (albeit mainly in Europeans), with many variants robustly replicated across tens of association studies. We sought to leverage this prior knowledge in order to empower our GWAS. Statistical theory indicates that incorporating known covariates in a linear regression model increases power to detect association[58], and simulation studies show that this applies to GWAS of population samples[59]. The situation in case–control studies of disease is more complex because in that setting association testing is affected by disease prevalence and effect sizes[59,60], so that disease GWASs have only occasionally conditioned on established loci[61]. However, conditioning on known large-effect SNPs in an unselected population sample (like the CANDELA cohort) for common pigmentation variation is an ideal setting in which to exploit the added power provided by conditional analyses. We thus examined which established pigmentation SNPs had strong effects in our sample

and used them to perform a conditioned GWAS. Searching online GWAS catalogs and published studies, we identified 161 SNPs that have been reported in previous association studies of pigmentation traits (Supplementary Table 12). Of these SNPs, 139 SNPs were present in the CANDELA imputed dataset (the rest being lost during QC). We obtained $P$ values and proportions of trait variance explained for each these 139 SNPs. We then selected SNPs that were both genome-wide significant ($P$ value < $5 \times 10^{-8}$) and that explained a relatively large proportion of trait variance (proportion of $R^2 > 0.5\%$, Supplementary Table 13) to define a list of established pigmentation SNPs with strong effects in the CANDELA sample. If several of these SNPs were located in the same gene region (usually a region with strong LD), and in order to avoid collinearity, we retained only the most significant SNP. The following six SNPs met these criteria and were used to perform a conditioned GWAS: rs16891982 (*SLC45A2*), rs12203592 (*IRF4*), rs10809826 (*TYRP1*), rs1800404 (*OCA2*), rs12913832 (*HERC2*) and rs1426654 (*SLC24A5*).

The polygenicity of the pigmentation traits examined in the CANDELA sample was evaluated using the tail strength (TS) statistic[25], which measures the overall strength of univariate (single-SNP) associations in a genome-wide test dataset. This statistic is related to other multiple-testing methods calculated on a set of $P$ values, like the false discovery rate and the area under the curve. In a GWAS with $n$ SNPs, if the ordered $P$ values are $p_{(1)} \leq p_{(2)} \leq \ldots \leq p_{(n)}$, the statistic is

$$TS(p_1, \ldots, p_n) = \frac{1}{n} \sum_{k=1}^{n} \left( 1 - p_k \frac{n+1}{k} \right). \qquad (1)$$

Under the null hypothesis of no association between the trait and all SNPs, TS should equal 0. A positive value of TS indicates the overall extent of association in the entire dataset and is interpreted as polygenicity, with higher values of TS indicating greater polygenicity. The asymptotic variance of TS can be approximated by is $1/n^*$ where $n^*$ is the effective number of independent SNPs. As LD pruning on our dataset yielded 160,858 SNPs (see Methods), the SD can be estimated as $1/\sqrt{160,858} = 0.0025$, and a confidence interval would be TS $\pm 3 \times$ SD = TS $\pm 0.0075$. The estimates of TS statistics obtained in the GWAS analyses performed in CANDELA data are shown in Supplementary Table 4A (and compared with the standard genomic inflation factor, $\lambda$). For three previously published GWAS studies on the same CANDELA cohort and using the same genetic PCs, lambda and TS statistic values are very close to zero for some traits that show few or no associations (Supplementary Table 4B), indicating that there is no inherent substructure remaining in the dataset after controlling with the genetic PCs. Results from other published GWAS studies show that lambda and TS values vary considerably within the same study, having highest values for pigmentation traits, height and body mass index, which have the largest number of associated SNPs (Supplementary Table 4C).

To evaluate association with all pigmentation traits simultaneously (excluding categorical eye color), we performed a Wald test[62]. In this approach, a SNP genotype is taken as the dependent variable and all phenotypes are jointly taken as covariates. Due to this increased complexity the runtime per SNP is considerably longer, so an LD-pruned dataset of 181,139 SNPs was used for this analysis (ensuring that all genome-wide and suggestive SNPs from the primary analysis are included) (Supplementary Table 6). A meta-analysis was carried out for the novel index SNPs identified in the primary analyses (Table 1) by testing for association separately in each country sample[16]. Forest plots were produced with MATLAB 3.2.5 combining all regression coefficients and standard errors. Histograms of the traits within each country were compared to the Forest plots to examine how trait variability across countries relates to the association signals.

**Review of functional annotation and gene expression data**. Functional annotation in the genomic regions showing association was reviewed using HaploReg v4.1[63], National Center for Biotechnology Information (NCBI), University of California Santa Cruz (UCSC) and Ensemble databases. Evolutionary constraint in these regions was assessed with the GERP[64] and SiPhy[65] scores. To evaluate the potential impact of amino-acid substitutions on protein structure and function, we examined the SIFT[66] and PolyPhen2[67] scores. We also queried transcription levels for candidate genes in newly associated regions across all 53 human tissues included in the GTEx database[68].

**Selection analyses**. We computed three selection statistics: the PBS[69], iHS[70] and Tajima's $D$[71]. Since we were mainly interested in the convergent evolution of pigmentation in West and East Eurasia, we restricted this analysis to CEU and CHB data from the 1000 Genomes Project. PBS scores for CEU were computed using CHB and YRI as reference and for CHB using CEU and YRI as reference. Pairwise $F_{ST}$ were estimated using Reynolds equation[72] using only SNPs that were polymorphic in at least two populations. The total number of SNPs with PBS scores in CHB and CEU was ~8,000,000. We calculated iHS using the software selscan[73]. Ancestral allele states were retrieved from information present in the 1000 Genomes data VCF files (AA (ancestral allele) field) and SNPs with no ancestral allele state were discarded. Unstandardized iHS scores were only estimated for SNPs when: (1) derived allele frequencies >5% and <95%; and (2) the Extended Haplotype Homozygosity (EHH) does not decay below 0.05 after an interval of 1 Mb. The standardized iHS scores were then computed by binning the SNPs by allele

frequencies and subtracting the mean and dividing by the standard deviation to obtain a final standardized statistic with a mean of 0 and variance of 1. The HapMap GRCh37 genetic map was used to obtain genetic distances between SNPs. The final total number of SNPs in CEU and CHB was ~3,000,000. We calculated Tajima's $D$ using VCFtools[74] on non-overlapping windows of 10 kb and discarded windows that contained less than 5 SNPs. The final total number of windows for CEU and CHB was ~266,000. We computed empirical $P$ values using an outlier approach by ranking all the genome-wide scores and dividing by the number of values in the distribution, taking the upper tail for PBS and iHS and the lower tail for Tajima's $D$ selection scores. Throughout the text we considered SNPs with significant selection scores as those with empirical $P$ values lower than 0.01.

To evaluate an enrichment of selection signals at genomic regions associated to pigmentation traits we first estimated haplotype blocks in the CANDELA sample using the definition of haplotype blocks implemented in PLINK 1.9[22]. When constructing haplotype blocks, only pair of SNPs within 500 Kb of each other were considered. For each haplotype block we then estimated the maximum PBS and iHS scores computed in the CEU and CHB populations, and retained only haplotype blocks with at least 5 SNPs. We then contrasted the distribution of maximum PBS and iHS scores at the haplotype blocks containing associated SNPs (i.e., those including SNPs with $P$ values < $10^{-5}$) with the distribution of maximum PBS and iHS scores at haplotype blocks in the rest of the genome. We tested the significance of the difference between distributions using a one-sided Mann–Whitney $U$-test. We did not use Tajima's $D$ selection scores to perform this enrichment analysis, as this selection statistic is computed in sliding windows (see above) and the windows would sometimes overlap two consecutive haplotype blocks.

To evaluate the possible correlation of allele frequencies at pigmentation genes with solar radiation levels we examined publicly available data for 64 native population samples without evidence of recent admixture (Supplementary Table 10). All samples included a minimum of 10 individuals. Surface solar radiation data were obtained from the NASA Surface meteorology and Solar Energy (SSE) Web site (https://eosweb.larc.nasa.gov/sse/) in kWh/m²/day units. These data included annual solar radiation averages from July 1983 to June 2005 on a 1-degree resolution grid over the globe. Annual solar radiation values were obtained for each population based on published coordinates for sampling locations. In case of unpublished sampling location, we obtained this information directly from the authors or used approximate coordinates such as the middle of the town/city of the sampling location. We used Bayenv2.0[75] to estimate Bayes Factors (BFs) relating solar radiation to allele frequencies at index SNP. These BFs provide a measure of the increase in the fit of allele frequencies to a linear regression model including solar radiation levels over a null model including only population structure as predictor. The null model was constructed using a covariance matrix of allele frequencies between populations estimated from 10,000 random SNPs (not in LD) after 100,000 Markov chain Monte Carlo iterations. In addition to BFs we estimated Spearman's rank correlation coefficient ($\rho$). We ranked the SNPs based on their BFs, and absolute $\rho$, to obtain empirical $P$ values. The allele frequency at a SNP was only considered to be significantly associated to solar radiation if both BF and $\rho$ estimates showed significance as recommended by Bayenv2.0. As the effect of pigmentation genes could differ between geographic regions, we also conducted separate analyses for Africans, Western Eurasians (including North Africans) and Eastern Eurasians (Supplementary Table 10 lists the populations included in each region).

To estimate the selection coefficient and the time since the start of selection at SNP (rs2240751), we used an ABC approach. We used msms[76] to perform coalescent simulations modeling the demographic history of African, European and East Asian populations (for details of the parameters of the demographic model used, see ref.[77] and Supplementary Note 1). We assumed that the minor allele frequency at the time of selection was 1% in Europeans and East Asians and zero in Africans (comparable to the frequency in CEU, CHB and YRI from the 1000 Genomes Project). We performed 1,000,000 simulations of a 500 kb genome segment with a selected allele in the center, and originating in East Asians. We assumed a uniform distribution U (0–0.05) for the selection coefficient and a uniform distribution U (5000–42,229 years ago (ya)) for the starting time of selection. From the simulations we computed 9 summary statistics in a window of 200 kb centered around the selected site: the nucleotide diversity ($\pi$), Tajima's $D$, Fu and Li's $D$, Fu and Li's $F$, H1, H2 and H2/H1 as measures of haplotype diversity, $F_{ST}$ between East Asians and Europeans, $F_{ST}$ between East Asians and Africans and the derived allele frequency of the selected variant. We used partial least squares (PLS) to identify the most informative statistics based on a subset of 10,000 simulations (prior to PLS analysis, summary statistics were Box-Cox transformed so that their minimum values were between 1 and 2). For parameter inference we used the first 7 PLS components, as they carried the most information for each parameter (estimated using the root mean squared error) (Supplementary Figure 14). Estimation of parameters was performed using the abc R package[78]. We selected the top 0.5% simulations based on the smallest Euclidean distance between the observed and simulated summary statistics. From these quantities, we obtained the posterior probability distributions for the selection coefficient and the time since selection, and recorded the posterior median and the 95% credible intervals. We examined the accuracy of the ABC parameter estimates using the predicted error (i.e., the mean square error divided by the prior variance of the parameter)

based on a leave-one-out cross-validation of 100 observations (Supplementary Figure 15).

Plots for the selection analyses were made in R.

**Immunohistochemistry of *MFSD12*.** Unshaven, full-thickness normal human adult scalp with terminal hair growth was used snap frozen in liquid nitrogen in cubes of 2 cm³. Cryosections of 6–8 μm were cut using a cryostat onto adhesive glass slides and stained with primary antibody against human C19Orf28/MFSD12 N-terminal region (MFSD1; Aviva System Biology ARP44958_P050) at a dilution of 1:600 using standard double immunofluorescence protocols. To assess the possible localization of MFSD12 in melanocytes of skin and/or hair follicles, we used a second primary antibody against the melanocyte lineage-specific antigen gp100. Quality testing of the antibody's specificity was assessed using commercially obtained sections of human kidney tissue as a positive control. IgG isotype controls were used at the same concentration as the lowest primary antibody dilution. Co-distribution and co-localization of both antigens in the skin and the growing hair follicle were determined if there was merging of the *MFSD12* (green)- and gp100 (red)-positive channels to give yellow/orange color. Human skin tissue used in this study was obtained with informed consent and with ethics committee approval.

**URLs.** For HaploReg, see https://pubs.broadinstitute.org/mammals/haploreg/haploreg.php. For NCBI, see https://www.ncbi.nlm.nih.gov. For UCSC, see https://genome.ucsc.edu. For Ensemble, see http://www.ensembl.org. For GTEx, see https://gtexportal.org/. For selscan, see https://github.com/szpiech/selscan.

## Data availability
Raw genotype or phenotype data cannot be made available due to restrictions imposed by the ethics approval. Summary statistics from the GWAS analyses is deposited at GWAS central with the link http://www.gwascentral.org/study/HGVST3308 (to be available upon next release in Spring 2019).

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

## Acknowledgements

We would like to dedicate this paper to Francisco M. Salzano. We thank the volunteers for their enthusiastic support for this research. We also thank Alvaro Alvarado, Mónica Ballesteros Romero, Ricardo Cebrecos, Miguel Ángel Contreras Sieck, Francisco de Ávila Becerril, Joyce De la Piedra, María Teresa Del Solar, Paola Everardo Martínez, William Flores, Martha Granados Riveros, Rosilene Paim, Ricardo Gunski, Sergeant João Felisberto Menezes Cavalheiro, Major Eugênio Correa de Souza Junior, Wendy Hart, Ilich Jafet Moreno, Paola León-Mimila, Francisco Quispealaya, Diana Rogel Diaz, Ruth Rojas, and Vanessa Sarabia for assistance with volunteer recruitment, sample processing and data entry. We also thank Richard Baker (Centre for Skin Sciences, University of Bradford) for technical assistance with the human skin immunofluorescence, Lewis Griffin (UCL Centre for Computer Science) for assistance in the development of iris color assessment and Emiliano Bellini for the face illustrations in Fig. 3. We also thank Louise Ormond and Aida Andres for helpful discussion on the ABC analysis (UCL Genetics Institute). We are very grateful to the institutions that allowed the use of their facilities for the assessment of volunteers, including: Escuela Nacional de Antropología e Historia and Universidad Nacional Autónoma de México (México); Universidade Federal do Rio Grande do Sul (Brazil); 13º Companhia de Comunicações Mecanizada do Exército Brasileiro (Brazil); Pontificia Universidad Católica del Perú, Universidad de Lima and Universidad Nacional Mayor de San Marcos (Perú). Work leading to this publication was funded by grants from: the Leverhulme Trust (F/07 134/DF), BBSRC (BB/I021213/1), the Excellence Initiative of Aix-Marseille University–A*MIDEX (a French 'Investissements d'Avenir' programme), Universidad de Antioquia (CODI sostenibilidad de grupos 2013–2014 and MASO 2013–2014), Consejo Nacional de Desenvolvimento Científico e Tecnológico, Fundação de Amparo à Pesquisa do Estado do Rio Grande do Sul (Apoio a Núcleos de Excelência Program) and Fundação de Aperfeiçoamento de Pessoal de Nível Superior. J.M.-R. was supported by a doctoral scholarship from CONCYTEC-PERU (224–2014-FONDECYT).

## Author contributions

K.A., J.M.-R., A.S., M.F.-G., J.L., J.C.C.-D. and D.J.T. performed the analyses. K.A., J.M.-R., D.J.T. and A.R.-L. wrote the paper with input from co-authors. M.F. and D.B. provided advice on study design and statistical analysis. All authors, namely, M.H., V.V., V.G., V.A.-A., C.J., W.A., R.B.L., P.E., J.G.-V., H.V.-R., C.C.S.C., T.H., V.R., L.S.-F., F.M.S., R.G.-J., M.-C.B., S.C.-Q., C.G., G.P., G.B., and F.R. contributed to volunteer recruitment or collection of data. A.R.-L. coordinated the study.

## Additional information

**Competing interests:** J.C.C.-D. was employed by Living DNA from October 2017 to November 2018. The remaining authors declare no competing interests.

Kaustubh Adhikari[1], Javier Mendoza-Revilla[1,2], Anood Sohail[3], Macarena Fuentes-Guajardo[1,4], Jodie Lampert[5], Juan Camilo Chacón-Duque [1], Malena Hurtado[2], Valeria Villegas[2], Vanessa Granja[2], Victor Acuña-Alonzo[1,6], Claudia Jaramillo[7], William Arias[7], Rodrigo Barquera Lozano[6,8], Paola Everardo[6], Jorge Gómez-Valdés[6], Hugo Villamil-Ramírez[9], Caio C. Silva de Cerqueira[10], Tábita Hunemeier[10], Virginia Ramallo[10,11], Lavinia Schuler-Faccini[10], Francisco M. Salzano[10], Rolando Gonzalez-José[11], Maria-Cátira Bortolini[10], Samuel Canizales-Quinteros[9], Carla Gallo[2], Giovanni Poletti[2], Gabriel Bedoya[7], Francisco Rothhammer[12,13], Desmond J. Tobin [14,15], Matteo Fumagalli[16], David Balding[1,17] & Andrés Ruiz-Linares[18,19]

[1]Department of Genetics, Evolution and Environment, and UCL Genetics Institute, University College London, London WC1E 6BT, UK. [2]Laboratorios de Investigación y Desarrollo, Facultad de Ciencias y Filosofía, Universidad Peruana Cayetano Heredia, Lima 31, Peru. [3]Department of Genetics, Cambridge University, Cambridge CB2 3EH, UK. [4]Departamento de Tecnología Médica, Facultad de Ciencias de la Salud, Universidad de Tarapacá, Arica 1000000, Chile. [5]Department of Genetics and Genome Biology, University of Leicester, Leicester LE1 7RH, UK. [6]National Institute of Anthropology and History, Mexico City 4510, Mexico. [7]GENMOL (Genética Molecular), Universidad de Antioquia, Medellín 5001000, Colombia. [8]Department of Archaeogenetics, Max Planck Institute for the Science of Human History, Jena 07745, Germany. [9]Unidad de Genomica de Poblaciones Aplicada a la Salud, Facultad de Química, UNAM-Instituto Nacional de Medicina Genómica, Mexico City 4510, Mexico. [10]Departamento de Genética, Universidade Federal do Rio Grande do Sul, Porto Alegre 91501-970, Brazil. [11]Instituto Patagonico de Ciencias Sociales y Humanas, Centro Nacional Patagonico, CONICET, Puerto Madryn U9129ACD, Argentina. [12]Instituto de Alta Investigación, Universidad de Tarapaca, Arica 1000000, Chile. [13]Programa de Genetica Humana, ICBM, Facultad de Medicina, Universidad de Chile, Santiago 8320000, Chile. [14]Centre for Skin Sciences, Faculty of Life Sciences, University of Bradford, Bradford BD7 1DP West Yorkshire, UK. [15]The Charles Institute of Dermatology, University College Dublin, Dublin D4, Ireland. [16]Department of Life Sciences, Silwood Park campus, Imperial College London, Ascot SL5 7PY, UK. [17]Melbourne Integrative Genomics, Schools of BioSciences and Mathematics & Statistics, University of Melbourne, Melbourne, VIC 3010, Australia. [18]Ministry of Education Key Laboratory of Contemporary Anthropology and Collaborative Innovation Center of Genetics and Development, School of Life Sciences and Human Phenome Institute, Fudan University, Shanghai 200438, China. [19]Aix-Marseille Université, CNRS, EFS, ADES, Marseille 13005, France. These authors contributed equally: Kaustubh Adhikari, Javier Mendoza-Revilla. Deceased: Francisco M. Salzano.

