## [Peer Review File · Nature Communications]

Reviewers' comments:

Reviewer #1 (Remarks to the Author):

This manuscript analyzes pigmentation phenotype and genome-wide SNP data in an extensive sample of Latin American individuals from several populations, with the goal of identifying candidate genes that influence pigmentation variation in Latin America. The analyses are rigorous and well-described, and the authors identify both previously-known candidates as well as some novel candidates. I found it to be an interesting study, and I have just a few minor comments for the authors to consider:

p.10, with reference to Fig. 4, please indicate which 3 SNPs have derived alleles that are not associated with lower pigmentation as that is not obvious to me.

p.17, the divergence time of 21,200 years ago for Europeans and Asians is from a rather old study and based on SNP data; more recent studies based on whole genome sequences estimate this divergence to be around 40,000 years ago (e.g. Malaspina et al. Nature 2016; Mallick et al. Nature 2016). While I would not ask the authors to redo the ABC analysis, they should mention what the effect of an older divergence time would be on the results of their ABC analysis (presumably it would allow for an older onset of selection on MFSD12 and hence a lower selection coefficient, as there would then be more time for the observed allele frequency changes to occur).

p.20, correlation between skin pigmentation and latitude in the Americas: how well has this actually been studied? Could the authors take their individuals with the least admixture and see if there is any correlation between MI and latitude?

p.21, with respect to intra-individual variation in MI, the authors provide the median value, but it would be useful to see the entire distribution of the intra-individual variation in MI values, maybe compared to the inter-individual variation.

p.24, the authors use about 670,000 SNP genotypes to impute genotypes at a total of over 9 million SNPs. As more than 90% of the data are thus imputed, it would be reassuring to have some indication of the accuracy of the imputation - e.g. randomly omit 1-5% of the actual genotype data, do the imputation, and then compare the imputed genotypes to the real genotypes for the data not used in the imputation.

p.25, the authors state that individuals with high East Asian ancestry, as estimated by ADMIXTURE analysis, were excluded, but the ADMIXTURE analysis in Supplementary Figure 5 does not include East Asian ancestry

p.29, the PBS score with CHB as the target population presumably used CEU and YRI as reference populations, not CHB and YRI.

Figure 7 legend, "organ/yellow" should presumably be "orange/yellow"

Supplementary

p.26, the phrase (add ref to table) appears.

p.27, please provide actual values of the percentage of the total variance explained by PC1 and PC2.

p.41, the ADMIXTURE plot lacks country labels.

Finally, there are numerous minor grammatical errors throughout the text – none so serious as to impede understanding, but the text would still benefit from careful attention by a professional editor.

Reviewer #2 (Remarks to the Author):

The authors have conducted a GWAS of pigmentation phenotypes in a very impressive dataset—

~6,000 individuals from 5 Central/South American countries for which they have detailed pigmentation phenotypes for skin, eyes, and hair. Because the majority of GWAS studies of pigmentation have been conducted in Europeans, this study is of particular importance because it helps shed light on potential convergent evolution of light skin in Europeans and Asians. They used a novel approach to study pigmentation hues of eyes which was particularly informative. They confirmed a number of loci previously identified as being associated with pigmentation phenotypes and identified several potential novel loci. While I think this study is important, I also have some major concerns which are described below.

My biggest concern is that they have merged "raw" GWAS results ($p < 10^{-8}$) with p values obtained when they condition on the top hits. They don't distinguish between these two in any of the tables or figures. This is misleading because when considering the "raw" GWAS results, only a few regions have significant associations, all of which were previously identified. I agree that doing the conditional analyses can help identify variants that don't reach genome-wide significance but they need to be completely transparent and prevent confusion by giving the original raw p value for every variant they identify as associated with the traits in every table and figure.

An additional major concern is that they do not have functional data to support their claims of particular variants being causal. They need to considerably tone down the claims of identifying particular variants as being causal (perhaps describing in the sup docs rather than the main text). An additional major concern is that they rely on imputation of genotypes using a SNP array that has very sparse coverage across the genome and has major ascertainment bias of SNPs common in Europeans. For example, they genotyped ~675,000 SNPs but impute >8,000,000 SNPs used for analyses! It is important to show how well the imputation worked, generally, but particularly important to show how well it works for SNPs which they suggest may be causal (they could have a sup table showing all associated SNPs, raw p values, adjusted p values, r^2 values for imputation). It would be more convincing if they could genotype those SNPs in a subset of their samples to show how well the imputation works.

Specific comments:

Abstract: They need to be cautious about claiming that loci in broad genomic regions are "novel". For example, MFSD12 was recently identified as being a gene that plays a role in pigmentation so the gene itself is not novel.

They say that pigmentation traits were moderately and significantly correlated. Which is it?

They state: 2B). "Categorical eye color was highly correlated with the quantitative L (Lightness) eye color variable ($r = -0.78$), but only moderately and minimally correlated with cos(H) (Hue) and C (Saturation) (r of 0.40 and -0.08, respectively)". An r^2 of -.08 is so minimal that it's not clear why they report it.

After using the top 6 PCs in their linear regression they show that there is still evidence for population stratification for skin pigmentation. They argue that this is because skin pigmentation is "highly polygenic". In fact, most studies have shown that skin pigmentation is not a highly polygenic trait compared to traits like height or blood pressure. Also, how did they account for relatedness among their samples? It would have been informative for them to use a linear mixed model method which uses a kinship matrix to see if that more adequately controls for structure.

I thought that the meta-analysis and the admixture mapping added very little to their results and should not have been highlighted (and certainly not in such detail) in the main text. The meta-analysis focuses on countries rather than ethnic groups and largely replicates what they had already found in the single SNP study. It was only informative to show that in Brazil where there is more European ancestry that there was stronger association with variants identified as associated with skin pigmentation in Europeans. The admixture analysis was very problematic because it uses a SNP array

that is not very dense and there is inherent error in the local ancestry inference and it's not clear how that might affect their results. The majority of their associations were the same but they had broader and less informative association peaks. They identified two novel associations but it's not clear to me if this could just be due to chance or due to the inherent error in the local ancestry inference results. For example, is the recombination rate different in those regions? Could that have impacted the local ancestry results? I don't believe that these results strengthen this paper and should either be removed or discussed only in the sup docs.

It's interesting that they observed multiple independent associations but I found the "jumping around" and merging of results from the GWAS of different pigmentation traits to be confusing here and throughout. It would be a lot more straightforward to talk about results for each separately and then talk about the ones that overlap across traits.

They state:

"Other evidence suggests that these eight independently associated SNPs are functional. Four occur in exons (three resulting in non-conservative amino-acid substitutions) and four are intronic. The exonic SNP encoding a synonymous substitution in exon 10 of OCA2 (rs1800404) is located in a conserved binding site for transcription factor YY1, which has been shown to regulate pigmentation in animal models 41. Amongst the four intronic SNPs, for one of them (rs12913832) there is experimental evidence indicating that it is involved in regulation of transcription of OCA242."

I have several concerns with this. First, the fact that four variants are intronic certainly does not show proof of causality. Even the ones that are non-synonymous are suggestive but not proven to be causal without additional functional data. The exonic SNP in exon 10 of OCA2 (rs1800404) was recently shown to be associated with alternative splicing in a study of African populations (Crawford et al. 2017). When they discuss the intronic SNPs, are they actually referring to intronic SNPs in HERC2 as implied by the paper they cite which showed that variants in non-coding regions of HERC2 are associated with expression of OCA2?

I am concerned about their results showing very high interactions amongst loci impacting pigmentation. This implies that they are not acting in an additive manner and contradicts prior studies. I would like to see a straightforward analysis showing how much of the phenotypic variation for each trait is accounted for by (1) all candidate causal SNPs identified in the study (2) each locus separately. If the percent of the phenotype for the individual loci added together is close to the total percent variance accounted for by all loci together, that would be consistent with these loci acting in an additive manner. I would want to see exactly how much more of the variation would be explained by epistatic interactions.

The section on prediction accuracy was long and confusing as written. They need to explain what this test is showing. For example, what does it mean that eye color showed the strongest prediction accuracy. Prediction accuracy for what? Again, I didn't feel like this section added much to the paper.

I have particular concerns about their claims for identifying novel causal genes/variants on chromosomes 10 and 19. At chromosome 10 the association peak is very large, over 400 kb. They claim that EMX2 is a likely causal gene because it's a transcription factor that plays a role in regulating expression of genes that play a role in melanocyte biology. But they have no functional data to support this claim. How many other loci are in this 400 kb region? Can you rule out that they aren't playing a role? Could the candidate loci in this region be regulating a gene(s) that are even farther away? How many other genes does EMX2 regulate? Is it specific to melanocytes? What evidence do they have that their index SNP is causal?

Regarding MFSD12, again, they see an association peak of 100kb. Looking at the peak in the sup docs, it's not clear why they would pick that isolated SNP when there are many others that show "clusters" of association in that region (thought the SNP they highlight has the lowest p value). The fact that it is a non-synonymous SNP is not sufficient to claim that it is causal. Because that is a region of high recombination, how well did the imputation analysis work in that region? The allele frequency distribution of their candidate causal SNP is interesting. But can they rule out that this isn't simply due to a new mutation arising in an Asian population on a haplotype background containing a different causal variant? It's an interesting observation and an interesting candidate causal SNP for future functional studies but I think they need to tone this down and express more caution in interpretation of their results. I also found the results of the presence of the protein in the scalp but not the hair follicles to be confusing. I would like to see more discussion of how they interpret that result and how it would be consistent or inconsistent with the recent study of a mouse knockout at this locus described in Crawford et al. 2017. Also, I advise avoiding the racial term "Caucasian" and instead refer to European ancestry.

I have the same concern with the novel association results for eye color. These span very broad association peaks and they have no functional data to support their claims of possible causal genes or variants. I also found the discussion of levels of expression based on GTEX studies to be unconvincing for most loci. Indeed, based on the GTEX results shown in the sup docs there was only one locus (WFDC5) where the gene was highly expressed specifically in skin compared to other tissues. For the other loci I could not see any obvious pattern. Just because a gene is expressed in skin cells isn't a strong argument, particularly if it's expressed in many cell types.

This doesn't warrant a paragraph: "In the supplement we discuss other genes of potential interest, located in the regions implicated by the secondary association analyses (Supplementary Figure 8)."

What do they mean by a broad pattern of high PBS scores in CEU at DSTYK? Importantly, if there is strong recent selection on certain loci (such as the variant at MFSD) as they claim, why haven't they done other tests of neutrality to see if there is a signature of selection such as Tajima's D, Fay and Wu's H, and EHH? I would want to at least see the results of these tests of neutrality and if they do not show a signature of selection, then I'd want them to explain why there is an inconsistency.

Regarding their analysis of a correlation between UV radiation and allele frequencies at MFSD12, I am concerned that this analysis is not appropriate for a small geographic region. Importantly, could their results simply reflect differences in ancestry among the populations from each region? How did they control for ancestry and drift in this analysis?

Where are the citations for these claims? "Furthermore, age estimates for many pigmentation associated SNPs predate the migration out of Africa of modern humans. It has therefore been hypothesized that the origin of these variants could predate the increase in pigmentation assumed to have occurred as a result of body hair loss during human evolution in Africa."

They should explain why they don't think they replicated some of the associations found in Europeans due to the fact that other studies were done in Northern Europeans. Do they mean to say that southern Europeans were the ancestors of the populations studied? They should explain that and give citations or evidence to support that claim.

They state: " In addition to allelic heterogeneity at these two genes here we identify the first East Asian-specific skin pigmentation locus (MFSD12)". MFSD12 is not an Asian specific pigmentation gene.

They state: "The IMPUTE2 genotype probabilities at each locus were converted into best-guess genotypes using PLINK v1.974. SNPs with proportion of samples with uncalled genotypes > 5% and minor allele frequency < 1% were excluded." Why didn't they use methods that take into account the uncertainty of the imputed variants? That would be much more appropriate in this case, particularly given the very low density SNP arrays they are using.

They estimated heritability using an additive model but elsewhere they claim that there are strong interactions between loci.

How many individuals of high African ancestry did they remove from their analysis and why use a 20% threshold?

The analysis of polygenicity is unclear as written and it's not clear how to interpret it—it's very vague and I don't find it particularly informative.

It's not clear to me why they seem to have done different associations based on different ancestries and inferred number of average ancestry blocks in the ancestry mapping approach—each individual will have a different percent ancestry.

In the Tables they must show the raw p values for all loci rather than showing p values calculated using different approaches. Also, they state: Although this marker (rs3795556) does not reach genome-wide significance for L, another marker in the region (rs16855186, in LD with rs3795556) reaches genome-wide significance.". So, why do they highlight the marker that doesn't show genome wide significance?

The composite manhattan plot was confusing for me—importantly, only the original p values for the full GWAS should be shown.

Figures 3, 4, and 5 and figure 8 D, E would be more appropriate for the sup docs in my opinion.

In sup figure 13 it would be helpful to have arrows pointing to the skin cells

PBS statistics shown in the sup docs were not convincing for the variants on chr 6, chr 10, chrom 11, chr 20, chr 22, which are below the black lines showing 99% threshold. At chromosome 19 there are some SNPs outside of the 99% threshold but they don't include the predicated causal variant which they focus on in the text.

Reviewer #3 (Remarks to the Author):

This manuscript reports a pigmentation GWAS in Latin Americans, leveraging three way admixture to gain insight not only into the biology of pigmentation, but also the evolutionary history of the trait.

The detailed phenotyping of such a large cohort is commendable and I think may be unprecedented.

However, the analytic approach is unvalidated by the citation that they use to justify it. The results therefore are not only difficult to interpret, but also are misleading. GWAS results are typically reported for single SNP associations, not conditioning on other SNPs. Table 1 consists of a conglomeration of traditional analytic results and results from their method, conditioning on a set of SNPs. The marginal results should be reported exclusively in Table 1. Secondary analyses should be

reported elsewhere.

It appears that when they used the standard analytic approach, they only identified five loci at genome-wide significance, all of which had been previously found in pigmentation GWAS. They then proceeded to conduct a second analysis, conditioning of these five loci, and this analysis identified an additional seven loci that exceeded the threshold for significance. They justify this analytic approach with reference to a method developed to identify additional independent associations within a locus, not to increase power to detect associations outside of the locus (Yang et al., 2012). Within the referenced paper, a statement in the introduction is made that "A more general and comprehensive strategy would be to perform a conditional analysis, starting with the top associated SNP, across the whole genome followed by a stepwise procedure of selecting additional SNPs, one by one, according to their conditional P values. Such a strategy would allow the discovery of more than two associated SNPs at a locus^{7,11}." I presume that this is the rationale that Adhikari et al. are using to justify their analytic approach, taken slightly out of context. Importantly, the Yang et al. paper is not testing that approach, it is simply a statement they make in the introduction. Furthermore, the two references to that statement (7,11) included one paper focused on identifying additional variants at established loci (ref 7) and the second paper (ref 11) explicitly states: "Stepwise conditional logistic regression is sensitive to missing data and subtle genotyping error, so we therefore desired an ultra-high-quality dataset. Markers were excluded from all sample collections for deviation from Hardy-Weinberg equilibrium in controls ($P < 0.0001$) and/or differential missingness in no-call genotypes between cases and controls ($P < 0.001$) in any of the seven collections. Finally, we required a per-SNP call rate of $>99.95\%$ (a maximum of 12 no-call genotypes from 24,269 samples per autosomal marker), generating a data set of 139,553 markers (of which all but 372 indels are SNPs)." Adhikari et al. fail to meet this high bar, for example, using imputed genotypes and not excluding SNPs with deviations from HWE. Therefore, their analytic approach is not justified and the results are uninterpretable.

Aside from this, there are additional concerns that should be addressed.

They provide insufficient references to previous pigmentation GWAS in the introduction, citing only two review papers published in 2009 and 2013, leaving out more recent publications of which there are at least three.

Multiple grammatical and use of language errors:

- the word underline is used twice, whereas I think underscore or highlight are more commonly used.
- "the frequency of the derived allele at MFSD12 is significantly correlated with lower solar radiation in East Asians" East Asians don't have lower solar radiation, they have lower exposure to solar radiation, or East Asia has lower solar radiation.
- several examples of "the" missing or unneeded "an".

Its not clear to me why admixture mapping would detect additional regions (3p22 and 4p12-q12) that were not detected in the primary SNP-based tests (Supplementary Table 6)." It would be nice to have an explanation in the discussion.

Tables should report p-values, not log-transformed p-values.

They state that the residual population stratification detected for skin pigmentation ($\lambda=1.11$) is due to polygenicity, but they detect a similar number of associations across the all the traits, suggesting they are all polygenic and yet these other traits don't have residual population stratification. It therefore seems that their explanation is insufficient.

Figure 7d has an unexplained white arrow.

1. On the conditional GWAS analysis:

It is apparent that the conditional GWAS analyses were not explained sufficiently in the previous version of our manuscript and this led to misunderstandings regarding our approach. Major points requiring clarification are the following:

- (i) Previous publications have robustly identified loci influencing pigmentation traits. Importantly, certain of these loci explain a large proportion of the population variation in pigmentation. For example in Beleza et al. (PLoS Genetics 2013, doi:10.1371/journal.pgen.1003372), 4 loci explain 35% of the variation in skin color (including 13% beyond what is explained by continental ancestry). This large proportion of explained variation is a feature quite peculiar to certain non-pathological physical appearance traits, like pigmentation variation in the general population. For disease association studies (or traits such as stature) this is certainly not the case. The loci that have been identified in those cases explain a very small proportion of trait variance (typically much less than 1%, or ORs less than 1.5). A lot has been made in the literature of this “missing heritability” feature of discoveries from most GWAS studies.
- (ii) Other than loci with large effects on pigmentation variation, several hundred genes involved in pigmentation have been identified in animal studies and some of these could have small effects on pigmentation variation in the general human population (for instance over a hundred associations are included in the NHGRI-EBI GWAS Catalogue, (www.ebi.ac.uk/gwas) (Supplementary Table 11).
- (iii) In cases where a few loci explain a large proportion of phenotypic variance, while many other loci have smaller contributions, statistical theory indicates that the regression models used in GWAS should gain in power by conditioning on established loci with large-effects. This increase in power is a general principle of regression analysis. We have provided a mathematical derivation (in Supplementary Note 1) showing that conditioning improves precision of the regression estimate and power in such cases. Considering such conditioning in GWAS studies of disease is usually not warranted, since only a small proportion of trait variation is explained by established loci, which goes some way to explain why the approach we used here is not generally considered in the literature (although has been used to search for independent signals at established loci). But our approach is based on classic statistical theory on regression analysis.
- (iv) As requested by the reviewers we now provide comparative results from conditioned and un-conditioned analyses (Supplementary Table 13). As expected, the results obtained are consistent across analyses and with the expected increase in power upon conditioning.
- (v) We have made the above considerations more explicit throughout the revised version of the manuscript. In particular, in the methods section we include the following text:

“Since there are a number of robustly replicated pigmentation associations reported in the literature, we sought to exploit this prior knowledge in order to increase the power of our GWAS analyses. We examined which of the previously reported pigmentation-associated

SNPs had strong effects in our sample and used these SNPs to condition the GWAS. Statistical principles dictate that the inclusion in regression models of predictors known to explain a significant amount of variation in the outcome variable increases the power to detect novel predictors (Supplementary Note 1, {Rao, 1973 #89}). Searching online GWAS catalogs and published studies we identified 161 SNPs that have been reported in previous association studies of pigmentation traits (Supplementary Table 11). Of these SNPs, 139 SNPs were present in the CANDELA imputed dataset (the rest being lost during QC). We tested for association to these 139 SNPs and obtained P-values and proportions of trait variance explained for each SNP. We then selected SNPs that were both genome-wide significant ($-\log P\text{-value} > 7.3$) and that explained a substantial proportion of trait variance (proportion of $R^2 > 0.5\%$) to define a list of previously reported pigmentation SNPs with strong effects in the CANDELA sample. If multiple SNPs were located in the same gene region (usually a region with strong LD), to avoid collinearity we retained only the most significant SNP. The following six SNPs met these criteria and were used to condition the GWAS: rs16891982 (SLC45A2), rs12203592 (IRF4), rs10809826 (TYRP1), rs1800404 (OCA2), rs12913832 (HERC2), rs1426654 (SLC24A5). For reference, unconditioned association P values and proportion of variance explained by the SNPs in Table 1 are shown in Supplementary Tables 12 and 13, respectively. As expected from an increase in power, the P values from the conditioned analyses (Table 1) are smaller than those seen in the unconditional analyses (Supplementary Tables 12 and 13), including previously reported pigmentation associated SNPs (rs1042602 in TYR, rs885479 in MC1R). Of the five novel associations reported in Table 1 (conditioned), three (rs11198112 in EMX, rs2240751 in MFSD12 and rs17422688 in WFDC5) are also genome-wide significant in the unconditioned analyses and two (rs3795556 in DSTYK and rs5756492 in MPST) are just below the threshold for genome-wide significance in the unconditional analyses.”

2. On the reliability of the imputed and genotyped data:

Several comments by the reviewers indicate that we did not provide sufficient details on the data QC steps we carried out, including for the imputation. We have now extended our manuscript so as to provide additional clarification and information (especially in the methods section). It is also important to note that the index SNPs in the five novel pigmentation-associated regions identified here were not imputed. They are present in the chip used for genotyping. This is now made explicit in the notes to Table 1.

The main additions to our description of the QC are the following:

- (i) We provide further information on the QC performed on the chip genotype data, including several metrics from the genotype calling software so as to exclude poorly genotyped SNPs, such as: “The SNP quality metrics generated from the GenCall algorithm in GenomeStudio were used for quality control. SNPs with low GenTrain score (<0.7), low Cluster Separation score (<0.3) or high heterozygosity values ($|\text{het. excess}| > 0.5$) were excluded.”
- (ii) Similarly, we provide additional details on the QC of the imputed data. Including:
 - a. Specifying that we used the ‘info’ metric to assess imputation quality (according to IMPUTE2, higher values “indicate that a SNP has been imputed with high certainty”), which we filter on. In the caption of Table 1, we indicate the imputed SNPs, and clarify that all imputed SNPs reported there had their ‘info’ metric to be > 0.975 , the median value being 0.993.

- b. We explain that we converted the IMPUTE2 genotype probabilities into most probably genotypes (with PLINK) using at a certainty threshold of >90%. This means that genotype calls below that high level of certainty are marked as missing data. Furthermore imputed SNPs with >5% missingness are removed. These two filters imply that for the retained imputed SNPs, the level of confidence is very high. In a previous GWAS (Adhikari et al. 2016, doi: 10.1038/ncomms11616) we ran the GWAS using both PLINK and SNPTEST (which uses the genotype probabilities instead of inferred hard genotype calls), and the results were the same. We therefore did not repeat the SNPTEST analysis here.
- c. Reviewer #2 suggests we assess the reliability of imputation by masking part of the chip genotyped data and subsequently imputing it. In fact, the imputation software used (IMPUTE2), performs such masking analysis. It removes a genotyped SNP from the data, imputes it using nearby SNPs, and assesses the concordance between the original genotypes and the imputed genotypes. SNPs having a low concordance score are removed from the analysis. We now explain this procedure in detail in the methods, including the fact that the median concordance values of the SNPs retained for the GWAs analyses was 0.99. In addition, in the caption of Table 1, we clarify that all genotyped SNPs reported on that table had ‘concordance’ metrics over 0.9 (and a median of 0.98).

REVIEWER #1

This manuscript analyzes pigmentation phenotype and genome-wide SNP data in an extensive sample of Latin American individuals from several populations, with the goal of identifying candidate genes that influence pigmentation variation in Latin America. The analyses are rigorous and well-described, and the authors identify both previously-known candidates as well as some novel candidates. I found it to be an interesting study, and I have just a few minor comments for the authors to consider:

p.10, with reference to Fig. 4, please indicate which 3 SNPs have derived alleles that are not associated with lower pigmentation as that is not obvious to me.

We now indicate on the main text, which 3 SNPs are not associated with lower pigmentation phenotypes.

p.17, the divergence time of 21,200 years ago for Europeans and Asians is from a rather old study and based on SNP data; more recent studies based on whole genome sequences estimate this divergence to be around 40,000 years ago (e.g. Malaspina et al. Nature 2016; Mallick et al. Nature 2016). While I would not ask the authors to redo the ABC analysis, they should mention what the effect of an older divergence time would be on the results of their ABC analysis (presumably it would allow for an older onset of selection on *MFSD12* and hence a lower selection coefficient, as there would then be more time for the observed allele frequency changes to occur).

Following this comment, we decided to redo the ABC analyses using a more recent demographic model (Jounganus et al. 2017), which estimated the divergence time between East Asians and Europeans to be ~42.3kya (similar to the estimates reported in the publications cited by the reviewer). Similar to our previous ABC analyses, we estimated a relatively small selection coefficient and an onset of selection on *MFSD12* in East Asians long after their split from Europeans.

Reference:

Jounganus et al. 2017. Inferring the Joint Demographic History of Multiple Populations: Beyond the Diffusion Approximation. GENETICS. 206-3:1549-1567; <https://doi.org/10.1534/genetics.117.200493>

p.20, correlation between skin pigmentation and latitude in the Americas: how well has this actually been studied? Could the authors take their individuals with the least admixture and see if there is any correlation between MI and latitude?

This is an interesting suggestion. Unfortunately, our data is not well suited for this analysis. Our sample was mostly collected in large urban centres that have received extensive immigration from surrounding regions over the years (see references below). Thus, the birthplaces of the individuals (even the ones with mostly Native American ancestry) will not match the levels of solar radiation exposure where their ancestors evolved.

References:

DANE. Direccion Nacional de Estadistica. Censo General 2005: Perfil Medellin Antioquia, http://www.dane.gov.co/files/censo2005/PERFIL_PDF_CG2005/05001T7T000.PDF (2005).

IBGE. Diretoria de Pesquisas, Coordenacao de Populacao e Indicadores Sociais, Estimativas da populacao residente. Porto Alegre. <http://cod.ibge.gov.br/28XRM> (2017).

INEGI. Instituto Nacional de Estadística y Geografía. Ciudad de México.
<http://www.beta.inegi.org.mx/app/areasgeograficas/?ag=09> (2014).

INEI. Instituto Nacional de Estadística e Informática. Perú: Migraciones internas 1993-2007.
https://www.inei.gob.pe/media/MenuRecursivo/publicaciones_digitales/Est/Lib0801/libro.pdf (2007).

p.21, with respect to intra-individual variation in MI, the authors provide the median value, but it would be useful to see the entire distribution of the intra-individual variation in MI values, maybe compared to the inter-individual variation.

We have added this information as Supplementary Figure 16.

p.24, the authors use about 670,000 SNP genotypes to impute genotypes at a total of over 9 million SNPs. As more than 90% of the data are thus imputed, it would be reassuring to have some indication of the accuracy of the imputation - e.g. randomly omit 1-5% of the actual genotype data, do the imputation, and then compare the imputed genotypes to the real genotypes for the data not used in the imputation.

Please see response above “On the reliability of the imputation and chip data”. We have in fact done the test suggested by the reviewer.

p.25, the authors state that individuals with high East Asian ancestry, as estimated by ADMIXTURE analysis, were excluded, but the ADMIXTURE analysis in Supplementary Figure 5 does not include East Asian ancestry

We have updated Supplementary Figure 5 adding the ADMIXTURE analysis at K=4 including the East Asian (CHB) reference panel. 185 samples had >5% East Asian ancestry and these were excluded from the GWAS analyses. This is now mentioned in Supplementary Figure 5.

p.29, the PBS score with CHB as the target population presumably used CEU and YRI as reference populations, not CHB and YRI.

This typo has been corrected.

Figure 7 legend, “organ/yellow” should presumably be “orange/yellow”

This typo has been corrected.

Supplementary

p.26, the phrase (add ref to table) appears.

We have fixed this sentence.

p.27, please provide actual values of the percentage of the total variance explained by PC1 and PC2.

We have added the total proportion of variance explained by PC1 and PC2 beneath Supplementary Figure 3C.

p.41, the ADMIXTURE plot lacks country labels.

Country labels have now been added to this ADMIXTURE plot.

Finally, there are numerous minor grammatical errors throughout the text – none so serious as to impede understanding, but the text would still benefit from careful attention by a professional editor.

We have proof-read the text extensively, correcting all language errors that we identified.

REVIEWER #2

The authors have conducted a GWAS of pigmentation phenotypes in a very impressive dataset—~6,000 individuals from 5 Central/South American countries for which they have detailed pigmentation phenotypes for skin, eyes, and hair. Because the majority of GWAS studies of pigmentation have been conducted in Europeans, this study is of particular importance because it helps shed light on potential convergent evolution of light skin in Europeans and Asians. They used a novel approach to study pigmentation hues of eyes which was particularly informative. They confirmed a number of loci previously identified as being associated with pigmentation phenotypes and identified several potential novel loci. While I think this study is important, I also have some major concerns which are described below.

My biggest concern is that they have merged “raw” GWAS results ($p < 10^{-8}$) with p values obtained when they condition on the top hits. They don’t distinguish between these two in any of the tables or figures. This is misleading because when considering the “raw” GWAS results, only a few regions have significant associations, all of which were previously identified. I agree that doing the conditional analyses can help identify variants that don’t reach genome-wide significance but they need to be completely transparent and prevent confusion by giving the original raw p value for every variant they identify as associated with the traits in every table and figure.

We now explain more fully our approach of conditioning on previously known pigmentation loci that account for a large proportion of the phenotypic variance in our sample. Please see response above “On the conditional GWAS analyses”.

An additional major concern is that they do not have functional data to support their claims of particular variants being causal. They need to considerably tone down the claims of identifying particular variants as being causal (perhaps describing in the sup docs rather than the main text).

We have toned down our comments on the possibility of the identified variants being functional. We haven’t entirely removed these comments in the main text as it is usual for GWAS studies to discuss candidate genes/variants when describing the association results. Commenting on such candidate variants (based on current knowledge) in the regions identified suggests avenues for follow-up work.

An additional major concern is that they rely on imputation of genotypes using a SNP array that has very sparse coverage across the genome and has major ascertainment bias of SNPs common in Europeans. For example, they genotyped ~675,000 SNPs but impute >8,000,000 SNPs used for analyses! It is important to show how well the imputation worked, generally, but particularly important to show how well it works for SNPs which they suggest may be causal (they could have a sup table showing all associated SNPs, raw p values, adjusted p values, r^2 values for imputation). It would be more convincing if they could genotype those SNPs in a subset of their samples to show how well the imputation works.

The chip used in our study (Illumina’s Omni Express) has been used in very many successful GWAS studies published by a number of different groups, including three papers from our group (Adhikari 2015, 2016a, 2016b). The informativeness of this chip for genome-wide imputation and the identification of trait loci has therefore been amply validated. The most recent Illumina chip (GSA) in fact has a somewhat lower SNP density (~610K) compared with the OmniExpress. We agree with the reviewer that confirming the reliability of the imputed

data prior to the association analyses is very important and we have done so extensively. Please see response above: “On the reliability of the imputation and chip data”. Note also that the pigmentation loci identified here include many associated SNPs, both genotyped and imputed. That is, evidence for association at those loci relies not only on the imputed data.

Specific comments:

Abstract: They need to be cautious about claiming that loci in broad genomic regions are “novel”. For example, *MFSD12* was recently identified as being a gene that plays a role in pigmentation so the gene itself is not novel.

The *MFSD12* locus was identified while our paper was being prepared for submission. We are therefore acutely aware of the Crawford et al. (2017) paper. We cite that paper in the results and discussion sections. However, a key fact is that our *association* is novel. That is, the top associated SNPs in our study differ entirely from those identified by Crawford et al (2017). Our index SNP is exonic and changes an amino acid in *MFSD12*, while the top SNPs detected in the Crawford et al paper are all intronic. The derived allele at our index SNP is seen at high frequency in East Asians and Native Americans but is absent from Africans. In contrast, the derived alleles at the SNPs detected in the Crawford paper are seen at high frequency mainly in Africans. The two associations signals detected in our papers are clearly different, albeit implicating the same gene. Since these points might have not have been clearly explained in the initial version of the manuscript we have now rephrased them in various parts of the current version, including the abstract.

They say that pigmentation traits were moderately and significantly correlated. Which is it? They state: 2B). “Categorical eye color was highly correlated with the quantitative L (Lightness) eye color variable ($r=-0.78$), but only moderately and minimally correlated with $\cos(H)$ (Hue) and C (Saturation) (r of 0.40 and -0.08 , respectively)”. An r^2 of -0.08 is so minimal that it’s not clear why they report it.

We reported the low correlation of categorical eye color with Hue and Saturation in order to highlight the large proportion of eye color variation that is not captured by conventional categorical eye color classification. We have now made this justification explicit.

After using the top 6 PCs in their linear regression they show that there is still evidence for population stratification for skin pigmentation. They argue that this is because skin pigmentation is “highly polygenic”. In fact, most studies have shown that skin pigmentation is not a highly polygenic trait compared to traits like height or blood pressure. Also, how did they account for relatedness among their samples? It would have been informative for them to use a linear mixed model method which uses a kinship matrix to see if that more adequately controls for structure.

In the text we argue that the higher genomic inflation factor for skin pigmentation is not due to residual population stratification but to a relatively high polygenicity, as indicated by the tail strength statistic. We agree with the reviewer that other traits (such as height) are considerably more polygenic than pigmentation, and do not show loci with relatively large effects (like seen for pigmentation). But this does not mean that pigmentation is not polygenic. Already about a dozen genes have been identified as having substantial effects on pigmentation variation in humans. In Supplementary Table 11 we list more than a hundred SNPs that have been reported to be associated with pigmentation in previous studies. Our point is simply that there are likely

to be a substantial number of additional loci with relatively smaller effects. Our results are consistent with a recent study by Martin et al (2017).

Regarding the use of mixed models, to account for individual relatedness, in a 2015 publication on our study sample, we showed that linear mixed models yielded similar results to those obtained using linear regression models with genetic PCs as covariates (Adhikari et al., 2015). We have clarified in the manuscript that we remove all related individuals (3rd degree relatives and higher). It has been argued mathematically that since both genetic PCs and LMMs use the same genetic kinship matrix, in absence of close kinship, population substructure captured by genetic PCs in standard linear regression GWAS provide equivalent results to LMM (Hoffman 2013).

References:

Martin et al. An Unexpectedly Complex Architecture for Skin Pigmentation in Africans. *Cell* 30;171(6):1340-1353.e14. (2017).

Adhikari et al. A genome-wide association study identifies multiple loci for variation in human ear morphology. *Nat Commun* 6, 7500 (2015).

Hoffman GE. Correcting for Population Structure and Kinship Using the Linear Mixed Model: Theory and Extensions. *PLoS ONE* 8(10): e75707 (2013).

I thought that the meta-analysis and the admixture mapping added very little to their results and should not have been highlighted (and certainly not in such detail) in the main text. The meta-analysis focuses on countries rather than ethnic groups and largely replicates what they had already found in the single SNP study. It was only informative to show that in Brazil where there is more European ancestry that there was stronger association with variants identified as associated with skin pigmentation in Europeans. The admixture analysis was very problematic because it uses a SNP array that is not very dense and there is inherent error in the local ancestry inference and it's not clear how that might affect their results. The majority of their associations were the same but they had broader and less informative association peaks. They identified two novel associations but it's not clear to me if this could just be due to chance or due to the inherent error in the local ancestry inference results. For example, is the recombination rate different in those regions? Could that have impacted the local ancestry results? I don't believe that these results strengthen this paper and should either be removed or discussed only in the sup docs.

Following this reviewer's suggestion we have moved the meta-analysis results to the supplement and removed altogether the admixture-mapping results.

It's interesting that they observed multiple independent associations but I found the "jumping around" and merging of results from the GWAS of different pigmentation traits to be confusing here and throughout. It would be a lot more straightforward to talk about results for each separately and then talk about the ones that overlap across traits.

We have tried to avoid the jumping around referred to by this reviewer and hope that the presentation is now clearer.

They state:

"Other evidence suggests that these eight independently associated SNPs are functional. Four occur in exons (three resulting in non-conservative amino-acid substitutions) and four are intronic. The exonic SNP encoding a synonymous substitution in exon 10 of OCA2 (rs1800404) is located in a conserved binding site for transcription factor YY1, which has been shown to regulate pigmentation in animal

models 41. Amongst the four intronic SNPs, for one of them (rs12913832) there is experimental evidence indicating that it is involved in regulation of transcription of OCA242.” I have several concerns with this. First, the fact that four variants are intronic certainly does not show proof of causality. Even the ones that are non-synonymous are suggestive but not proven to be causal without additional functional data. The exonic SNP in exon 10 of OCA2 (rs1800404) was recently shown to be associated with alternative splicing in a study of African populations (Crawford et al. 2017). When they discuss the intronic SNPs, are they actually referring to intronic SNPs in HERC2 as implied by the paper they cite which showed that variants in non-coding regions of HERC2 are associated with expression of OCA2?

We agree with the reviewer that the functional annotation available for the associated variants is only suggestive and does not prove causality. However, we believe mentioning the functional annotations in the associated regions identified provides important information allowing the reader to contextualize our findings. We have therefore carefully reworded this section, including making explicit which specific SNPs we are referring to when providing genome annotation information.

I am concerned about their results showing very high interactions amongst loci impacting pigmentation. This implies that they are not acting in an additive manner and contradicts prior studies. I would like to see a straightforward analysis showing how much of the phenotypic variation for each trait is accounted for by (1) all candidate causal SNPs identified in the study (2) each locus separately. If the percent of the phenotype for the individual loci added together is close to the total percent variance accounted for by all loci together, that would be consistent with these loci acting in an additive manner. I would want to see exactly how much more of the variation would be explained by epistatic interactions.

Following the reviewer’s suggestion we calculated what fraction of the trait variance (measured as % R²) are explained by various factors. To avoid overfitting, this was estimated through cross-validation following the prediction procedure. As we have now removed the prediction section based on the reviewer’s suggestion, we are showing the table here.

Factor	Skin	Hair	Eye			
	MI	Categorical	Categorical	L (Brightness)	C (Saturation)	cos(H) (Hue)
Ancestry (continental)	24.2	14.9	16.6	22.3	5.4	4.2
Genetic PCs	26.5	16.8	19.4	24.2	16.1	10.9
Index SNPs (main effects)	18.1	12.3	30.1	33.6	6.7	7.8
Interaction of Index SNPs (LASSO)	0.1	5	5.9	1.2	4	9.5
Interaction of Index SNPs (Random Forests)	6.7	6.6	1.1	2.1	5.8	19

For melanin index, proportion of trait variance explained by genetic PCs is 27% and by all index SNPs (beyond PCs) is 18%. Other published studies of pigmentation have observed similar values, e.g. Belezza et al. (2013) on admixed Cape Verde populations noted 44% of skin color variation being explained by ancestry and 13% by four index SNPs beyond ancestry, while Martin et al. (2017) note 34% of variation in skin color being explained by ancestry in African KhoeSan.

The proportion of trait variance explained by interactions is generally smaller than that explained by main effects, as expected, but is considerable for some traits, being up to 7%, though $\cos(H)$ had much higher values (10-20%). Traits that have high R^2 values for interaction under LASSO (a linear model) generally show more significant interactions in Figure 4 (also obtained using linear regression).

Reference:

Beleza, S. et al. Genetic architecture of skin and eye color in an African-European admixed population. PLoS Genet 9, e1003372 (2013).

Martin, A.R. et al. An Unexpectedly Complex Architecture for Skin Pigmentation in Africans. Cell 171, 1340-1353 e14 (2017).

The section on prediction accuracy was long and confusing as written. They need to explain what this test is showing. For example, what does it mean that eye color showed the strongest prediction accuracy. Prediction accuracy for what? Again, I didn't feel like this section added much to the paper.

Following this reviewer's suggestion we have deleted the section on prediction from the manuscript.

I have particular concerns about their claims for identifying novel causal genes/variants on chromosomes 10 and 19. At chromosome 10 the association peak is very large, over 400 kb. They claim that EMX2 is a likely causal gene because it's a transcription factor that plays a role in regulating expression of genes that play a role in melanocyte biology. But they have no functional data to support this claim. How many other loci are in this 400 kb region? Can you rule out that they aren't playing a role? Could the candidate loci in this region be regulating a gene(s) that are even farther away? How many other genes does EMX2 regulate? Is it specific to melanocytes? What evidence do they have that their index SNP is causal?

We agree with the reviewer that the Chr. 10 region is a large and there is no overwhelming evidence implicating a specific functional variant in that region. Our intention in that section is to comment on available information on candidate genes/variants in the region. We have rephrased this section, hopefully making it clearer that we do not intend to provide proof regarding a specific causal variant, but only providing contextual information on the region, based on available annotations.

Regarding MFSD12, again, they see an association peak of 100kb. Looking at the peak in the sup docs, it's not clear why they would pick that isolated SNP when there are many others that show "clusters" of association in that region (thought the SNP they highlight has the lowest p value). The fact that it is a non-synonymous SNP is not sufficient to claim that it is causal. Because that is a region of high recombination, how well did the imputation analysis work in that region? The allele frequency distribution of their candidate causal SNP is interesting. But can they rule out that this isn't simply due to a new mutation arising in an Asian population on a haplotype background containing a different causal variant? It's an interesting observation and an interesting candidate causal SNP for future functional studies but I think they need to tone this down and express more caution in interpretation of their results. I also found the results of the presence of the protein in the scalp but not the hair follicles to be confusing. I would like to see more discussion of how they interpret that result and how it would be consistent or inconsistent with the recent study of a mouse knockout at this locus described in Crawford et al. 2017. Also, I advise avoiding the racial term "Caucasian" and instead refer to European ancestry.

I have the same concern with the novel association results for eye color. These span very broad association peaks and they have no functional data to support their claims of possible causal genes or variants. I also found the discussion of levels of expression based on GTEX studies to be unconvincing for most loci. Indeed, based on the GTEX results shown in the sup docs there was only one locus (WFDC5) where the gene was highly expressed specifically in skin compared to other tissues. For the other loci I could not see any obvious pattern. Just because a gene is expressed in skin cells isn't a strong argument, particularly if it's expressed in many cell types.

We agree with the reviewer that there is no overwhelming evidence implicating a specific, well-established pigmentation variant in the MFSD12 gene region. This is the usual situation in all novel discoveries made through GWAS analyses. When new associations are identified they usually happen at loci for which there is no prior overwhelming evidence implicating them. In many ways, that is what makes the finding novel. However, contextual information helps make the case for specific variants having a greater chance of being causal and providing a biological explanation to the association. This includes whether a particular variant is in a functional domain of a protein, or changes an amino acid. Another is the strength of the P-value. In very many previous GWAS studies the SNP with the strongest association is likely to represent, or be very close to, the functional variant underlying the association. That appears to be the case at the other amino-acid changing variants listed in Table 1) detected at other now well-established pigmentation genes. As argued above, we are not trying to make absolute claims regarding the functionality of this locus, but only providing contextual information based on the available annotations. We have rephrased this section, hopefully making our assertions clearer.

The index SNP at MFSD12 was chip genotyped, passing all QC thresholds, and having a 'concordance' metric (for genotyping accuracy) of >0.9 from IMPUTE2. For details, please see response above: "On the reliability of the imputation and chip data".

Regarding the significance of our results in the light of the recent Mfsd12 mouse mutant Crawford et al (2017), we believe it is difficult to compare directly the impact of CRISPR-Cas9 cleavage resulting in a null allele of Mfsd12 gene in the mouse in with our reported expression of MFSD12 in normal human haired skin. This for several reasons.

Firstly, Crawford et al does not report on the expression of Mfsd12 protein in the normal littermates or how a null allele of Mfsd12 gene may affect its protein expression generally in the skin.

Secondly, melanocytes are not present in mouse pelage skin (epidermis) but are restricted to their hair follicle, in marked contrast to human skin where pigment cells are present in both skin (epidermis) and hair follicles (see: Tobin DJ (2010) The cell biology of human hair follicle pigmentation. *Pigment Cell Melanoma Res.* 2011 Feb;24(1):75-88).

Thirdly, melanocytes in human epidermis and hair follicle are distinct and regulated very differently (Tobin DJ, Bystryrn JC. Different populations of melanocytes are present in hair follicles and epidermis. *Pigment Cell Res.* 1996;9(6):304-10).

It is of particular interest in the current study that MFSD12 protein expression was readily detected in melanocytes of the epidermis but not of the hair follicle, and that this concurs with our reported observations on associated skin, not hair, pigmentation in the cohort studies in this study.

It is most likely that selective pressures on hair versus skin color will have been significantly different during evolution, given that the epidermis (not the hair follicle) is the major source of vitamin D for the body. Moreover, ultraviolet radiation regulates pigmentation in melanocytes

of the epidermis, not the hair follicle, again suggesting different evolutionary selective pressures on gene expression in these respective pigmentary units. The precise function or mechanism of action of MFSD12 in human skin physiology (both for melanocytes themselves, and other cell types expressing this lysosomal protein including keratinocytes) remains to be determined.

We have replaced the term “Caucasian”, as suggested by the reviewer.

This doesn't warrant a paragraph: “In the supplement we discuss other genes of potential interest, located in the regions implicated by the secondary association analyses (Supplementary Figure 8).”

This sentence has been removed.

What do they mean by a broad pattern of high PBS scores in CEU at DSTYK? Importantly, if there is strong recent selection on certain loci (such as the variant at MFSD) as they claim, why haven't they done other tests of neutrality to see if there is a signature of selection such as Tajima's D, Fay and Wu's H, and EHH? I would want to at least see the results of these tests of neutrality and if they do not show a signature of selection, then I'd want them to explain why there is an inconsistency.

Following the reviewers' suggestion, we have computed two additional selection statistics: Taima's D and the integrated Haplotype Score (iHS), which generally are consistent with the PBS analyses. These additional results are now discussed in the text.

Regarding their analysis of a correlation between UV radiation and allele frequencies at MFSD12, I am concerned that this analysis is not appropriate for a small geographic region. Importantly, could their results simply reflect differences in ancestry among the populations from each region? How did they control for ancestry and drift in this analysis?

Considering this comment, we decided to redo entirely the geographic analyses using the Bayenv2.0 software, which incorporates a Bayesian model to test for correlations between allele frequencies and environmental variables. For each SNP Bayenv2.0 (Gunther & Coop, 2013) produces a Bayes Factor (BF) that measures the increase in the fit of a model with a linear relationship between allele frequencies and solar radiation over a null model in which the allele frequencies is dependent on population structure alone. As done in all previously published studies (and in our initial manuscript) we conducted this analysis on a worldwide dataset as well as performing separate analyses for populations from Western and Eastern Eurasia. The justification for performing regional analyses is that selection effects could be acting on specific local variants (i.e. those with a restricted geographic distribution) and therefore one would not expect to find a significant correlation in areas where the selected SNP is not segregating. We note that in the paper introducing the Bayenv method (Coop et al. 2010) the authors made this point explicit: “Methods based on environmental correlations will fail to detect such cases, unless the data are split into the appropriate geographic subsets (e.g., Hancock et al. 2011c) on an appropriate geographic scale (Ralph and Coop 2010).”

Similar to our initial analyses, the new BF analyses demonstrate a significant correlation of the MFSD12 variant and solar radiation in the Eastern Eurasian dataset.

References:

Gunther & Coop 2010 Robust Identification of Local Adaptation from Allele Frequencies. Genetics 2013:195-1 205 220; doi.org/10.1534/genetics.113.152462

Coop et al. 2010 Using Environmental Correlations to Identify Loci Underlying Local Adaptation. *Genetics*. 2010 Aug; 185(4): 1411–1423; doi: 10.1534/genetics.110.114819

Where are the citations for these claims? “Furthermore, age estimates for many pigmentation associated SNPs predate the migration out of Africa of modern humans. It has therefore been hypothesized that the origin of these variants could predate the increase in pigmentation assumed to have occurred as a result of body hair loss during human evolution in Africa.”

We thank the reviewer for pointing this out. We have added the missing references.

They should explain why they don't think they replicated some of the associations found in Europeans due to the fact that other studies were done in Northern Europeans. Do they mean to say that southern Europeans were the ancestors of the populations studied? They should explain that and give citations or evidence to support that claim.

Yes, that is exactly the case. The European ancestors of Latin Americans mostly stemmed from the Iberian peninsula. It is a well-documented historical fact that the countries sampled here were colonized by Spain or Portugal. We have added references to support this statement.

They state: “ In addition to allelic heterogeneity at these two genes here we identify the first East Asian-specific skin pigmentation locus (MFSD12).” MFSD12 is not an Asian specific pigmentation gene.

We have modified this sentence to clarify that the “novel association” refers to Asian-specific alleles at this gene.

They state: “The IMPUTE2 genotype probabilities at each locus were converted into best-guess genotypes using PLINK v1.974. SNPs with proportion of samples with uncalled genotypes > 5% and minor allele frequency < 1% were excluded.” Why didn't they use methods that take into account the uncertainty of the imputed variants? That would be much more appropriate in this case, particularly given the very low density SNP arrays they are using.

As explained above in the section entitled: “On the reliability of the imputation and chip data” we have previously performed analyses using genotype probabilities and obtained the same results as when using called genotypes (called using appropriate quality control filters, as explained above).

They estimated heritability using an additive model but elsewhere they claim that there are strong interactions between loci.

It is indeed a limitation that narrow-sense heritability (estimated using the GRM) only use an additive model. Though we now show the proportion of trait variance explained through interaction of the index SNPs also. We don't know of a reliable way of calculating broad-sense heritability (including all SNP interaction terms) from genome-wide data of unrelated individuals.

How many individuals of high African ancestry did they remove from their analysis and why use a 20% threshold?

A total of 188 individuals with high African ancestry were removed. By examining the long thin tail of the distribution of individual African ancestry values, we decided to use the upper 2.5%

quantile as the threshold. This equates to 20% ancestry. This is now mentioned in Supplementary Figure 5.

The analysis of polygenicity is unclear as written and it's not clear how to interpret it—it's very vague and I don't find it particularly informative.

We have extensively updated the text on polygenicity to provide further clarifications and results, including new supplementary tables 4B-C.

It's not clear to me why they seem to have done different associations based on different ancestries and inferred number of average ancestry blocks in the ancestry mapping approach—each individual will have a different percent ancestry.

As mentioned above, the admixture-mapping analysis has now been removed from the paper.

In the Tables they must show the raw p values for all loci rather than showing p values calculated using different approaches. Also, they state: Although this marker (rs3795556) does not reach genome-wide significance for L, another marker in the region (rs16855186, in LD with rs3795556) reaches genome-wide significance.” So, why do they highlight the marker that doesn't show genome wide significance?

We explain above that most of our analysis was conducted only on the conditioned GWAS analysis, and thus Table 1 mentions conditioned P-values for all SNPs other than the conditioning SNPs.

Signals of association were seen in that genetic region for many SNPs (in LD), and while the pattern of association was similar for both L and C, the most significant SNP was slightly different for the two traits. So, considering that this is a single genetic region and the two index SNPs are in LD, we preferred to simplify the table by reporting only one index SNP per LD block (i.e. per genetic region). Thus we presented as one row results for the SNP that was genome-wide significant for C, and added the result for the other SNP for L as it is the most significant for L.

The composite manhattan plot was confusing for me—importantly, only the original p values for the full GWAS should be shown.

We have expanded the legend of Figure 2 to explain more fully that composite Manhattan plots are used as a way to simplify display of the main results across the multiple GWAS performed. Otherwise we would have had to show 6 different Manhattan plots, something not viable in the main text. We feel it is informative for the reader to include a single Manhattan plot on which we overlay all the SNPs exceeding the genome-wide suggestive and significant thresholds. In this way the reader can appreciate at a glance (and in the main text) the major findings across all the GWAS performed. Using such composite Manhattan plots to represent multiple GWAS in a single figure is not an uncommon practice (see references below). We emphasize that this is simply a summary display tool, with the scientific interpretation of results being developed in the main text of the paper.

References:

- **Pallares LF, Carbonetto P, Gopalakrishnan S, Parker CC, Ackert-Bicknell CL, Palmer AA, et al. (2015) Mapping of Craniofacial Traits in Outbred Mice Identifies Major**

Developmental Genes Involved in Shape Determination. PLoS Genet 11(11): e1005607. doi:10.1371/journal.pgen.1005607

- **Nalls et al. (2014). Large-scale meta-analysis of genome-wide association data identifies six new risk loci for Parkinson's disease. Nature Genetics 46, 989–993 (2014). doi:10.1038/ng.3043**
- **Schunkert et al. (2011). Large-scale association analysis identifies 13 new susceptibility loci for coronary artery disease. Nature Genetics 43, 333–338 (2011). doi:10.1038/ng.784**
- **Chen et al. (2014). Genome-wide association analyses provide genetic and biochemical insights into natural variation in rice metabolism. Nature Genetics 46, 714–721 (2014). doi:10.1038/ng.3007**
- **Adhikari et al. (2015) A genome-wide association study identifies multiple loci for variation in human ear morphology. Nat Commun 6, 7500 (2015).**
- **Adhikari & Fuentes-Guajardo et al. (2016) A genome-wide association scan implicates DCHS2, RUNX2, GLI3, PAX1 and EDAR in human facial variation. Nat Commun 19;7:11616. doi: 10.1038/ncomms11616.**

Figures 3, 4, and 5 and figure 8 D, E would be more appropriate for the sup docs in my opinion.

Following this reviewer's suggestion, we have moved Figure 3, 8D and 8E to the Supplementary documentation.

In sup figure 13 it would be helpful to have arrows pointing to the skin cells

We have added arrows pointing to the skin cells in the Supplementary Figure.

PBS statistics shown in the sup docs were not convincing for the variants on chr 6, chr 10, chrom 11, chr 20, chr 22 , which are below the black lines showing 99% threshold. At chromosome 19 there are some SNPs outside of the 99% threshold but they don't include the predicated causal variant which they focus on in the text.

We are aware that the PBS score for the index SNP in *MFSD12* is not outside of the 99th percentile. It is however >98th percentile, with an overall enrichment of high PBS scores in the region (in the Supplementary documentation we have added a table showing the empirical rank for the scores obtained for the selection statistics). It is worth pointing out that these selection tests have not been proposed as approaches to identify the exact location of the variant driving selection in a region under selection.

REVIEWER #3

This manuscript reports a pigmentation GWAS in Latin Americans, leveraging three way admixture to gain insight not only into the biology of pigmentation, but also the evolutionary history of the trait.

The detailed phenotyping of such a large cohort is commendable and I think may be unprecedented.

However, the analytic approach is unvalidated by the citation that they use to justify it. The results therefore are not only difficult to interpret, but also are misleading. GWAS results are typically reported for single SNP associations, not conditioning on other SNPs. Table 1 consists of a conglomeration of traditional analytic results and results from their method, conditioning on a set of SNPs. The marginal results should be reported exclusively in Table 1. Secondary analyses should be reported elsewhere. It appears that when they used the standard analytic approach, they only identified five loci at genome-wide significance, all of which had been previously found in pigmentation GWAS. They then proceeded to conduct a second analysis, conditioning of these five loci, and this analysis identified an additional seven loci that exceeded the threshold for significance. They justify this analytic approach with reference to a method developed to identify additional independent associations within a locus, not to increase power to detect associations outside of the locus (Yang et al., 2012). Within the referenced paper, a statement in the introduction is made that “A more general and comprehensive strategy would be to perform a conditional analysis, starting with the top associated SNP, across the whole genome followed by a stepwise procedure of selecting additional SNPs, one by one, according to their conditional P values. Such a strategy would allow the discovery of more than two associated SNPs at a locus^{7,11}.” I presume that this is the rationale that Adhikari et al. are using to justify their analytic approach, taken slightly out of context. Importantly, the Yang et al. paper is not testing that approach, it is simply a statement they make in the introduction. Furthermore, the two references to that statement (7,11) included one paper focused on identifying additional variants at established loci (ref 7) and the second paper (ref 11) explicitly states: “Stepwise conditional logistic regression is sensitive to missing data and subtle genotyping error, so we therefore desired an ultra-high-quality dataset. Markers were excluded from all sample collections for deviation from Hardy-Weinberg equilibrium in controls ($P < 0.0001$) and/or differential missingness in no-call genotypes between cases and controls ($P < 0.001$) in any of the seven collections. Finally, we required a per-SNP call rate of $>99.95\%$ (a maximum of 12 no-call genotypes from 24,269 samples per autosomal marker), generating a data set of 139,553 markers (of which all but 372 indels are SNPs).” Adhikari et al. fail to meet this high bar, for example, using imputed genotypes and not excluding SNPs with deviations from HWE. Therefore, their analytic approach is not justified and the results are uninterpretable.

Please see response above “On the conditional GWAS analyses” and “On the reliability of the imputed and genotyped data”.

Aside from this, there are additional concerns that should be addressed.

They provide insufficient references to previous pigmentation GWAS in the introduction, citing only two review papers published in 2009 and 2013, leaving out more recent publications of which there are at least three.

We have added more recent citations to our introductory paragraph. We also provide a more exhaustive list of previous GWAS studies from the NHGRI-EBI GWAS Catalogue and other sources in Supplementary Table 11.

Multiple grammatical and use of language errors:

-the word underline is used twice, whereas I think underscore or highlight are more commonly used.

We have copy-edited the manuscript throughout.

-"the frequency of the derived allele at MFSD12 is significantly correlated with lower solar radiation in East Asians" East Asians don't have lower solar radiation, they have lower exposure to solar radiation, or East Asia has lower solar radiation.

We have changed this sentence to: “*We document that the frequency of the derived allele at MFSD12 is significantly correlated with lower exposure to solar radiation in East Asia...*”

-several examples of "the" missing or unneeded "an".

We have revised the manuscript extensively and correct these and other grammatical errors.

Its not clear to me why admixture mapping would detect additional regions (3p22 and 4p12-q12) that were not detected in the primary SNP-based tests (Supplementary Table 6).” It would be nice to have an explanation in the discussion.

As requested by another reviewer, we have now removed the admixture analysis results.

Tables should report p-values, not log-transformed p-values.

We respectfully disagree. Negative log P-values are a better way to convey the information in a GWAS. The exponent of the P-value is what really matters and that is directly provided by the log transformation. This is fairly standard practice in GWAS studies.

They state that the residual population stratification detected for skin pigmentation ($\lambda=1.11$) is due to polygenicity, but they detect a similar number of associations across the all the traits, suggesting they are all polygenic and yet these other traits don't have residual population stratification. It therefore seems that their explanation is insufficient.

We have improved the text in the manuscript to clarify that higher values of lambda for skin pigmentation are likely to be due to greater polygenicity and not to residual population stratification. Our argument is in fact the same being made by the reviewer: if there was residual population stratification, then that would cause an inflation of lambda for all traits, which is not the case. The number of index SNPs reported in Table 1 is highest for skin pigmentation, which is thought to more polygenic than eye or hair pigmentation, and has the TS value in our data. But as the reviewer suggests all pigmentation traits have some degree of polygenicity - over a hundred associations are included in the NHGRI-EBI GWAS Catalogue ([/www.ebi.ac.uk/gwas](http://www.ebi.ac.uk/gwas), Supplementary Table 11).

To support our claim that higher values of lambda and TS is due to polygenicity and not due to residual population stratification, we provide Supplementary Tables 4B-C showing lambda, TS and the number of significant associations for various phenotypes (examined in the CANDELA GWAS and other cohorts). We show that in the CANDELA cohort (using the same genetic PCs to correct for stratification), lambda and TS statistic values are very close to zero for traits that show few or no significant associations, indicating that there is no inherent substructure remaining in the dataset after controlling with the genetic PCs. Table 4C also shows that for

other traits lambda and TS values can vary considerably within the same study sample, and reach highest values for pigmentation, height and BMI, traits which have the largest number of associated SNPs (i.e. greatest polygenicity).

Figure 7d has an unexplained white arrow.

We have reworded the legend of this figure.

Reviewers' comments:

Reviewer #1 (Remarks to the Author):

The authors have responded satisfactorily to my comments and I have no further comments.

Reviewer #2 (Remarks to the Author):

This is a resubmission of a manuscript describing genome-wide associations with pigmentation phenotypes in a large sample of >6,000 Latin Americans. As noted in my previous review, this is an impressive dataset which adds important information about the genetics and evolution of pigmentation phenotypes in a region of the world which has not previously been well characterized. The authors have generally done a good job addressing the concerns of reviewers and I commend them for making this revised version much more streamlined and focused. It reads much better now (though there are still grammatical errors which can be corrected by an editor).

However, I still have a few serious concerns which can be easily addressed by the authors. The most important concern that I have is that their use of a "conditional" GWAS is still not warranted. This is an approach which has never been used in prior GWAS—they simply cite books on regression analysis for support. I am not convinced that this is an appropriate approach and it should be peer reviewed as a statistical methods paper rather than introduced as an "ad hoc" approach justified in a few paragraphs in the sup docs. It adds a layer of confusion that is simply not necessary (particularly as they go back and forth between "conditional" and "unconditional" GWAS results throughout the text and sup docs). They state "Of the five novel associations reported in Table 1 (conditioned), three (rs11198112 in EMX, rs2240751 in MFSD12 and rs17422688 in WFDC5) are also genome-wide significant in the unconditioned analyses and two (rs3795556 in DSTYK and rs5756492 in MPST) are just below the threshold for genome-wide significance in the unconditional analyses." So, if this is the case, they can simply state the results for the standard unconditional analyses in Table 1. They can still mention the associations that are just below threshold of genome-wide significance but they need to present them in a way such that they can be directly compared to other GWAS of pigmentation traits. Presenting the unconditioned results does not detract in any way from their paper but using an unconventional approach to do a "conditional GWAS" makes it extremely difficult to interpret and to compare to other studies. I feel strongly that it should not be the primary results described in the main text or in Table 1. I also agree with the third reviewer that they should not present their p values based on log-transformed p-values. While the authors prefer not to do this, it makes their results VERY difficult to interpret and to compare with other studies. Most readers will not be able to distinguish how strong the associations are compared to standard p values. There is simply no good reason not to list the raw p values here and in the Supp tables.

I have some other minor concerns that should be addressed:

Lines 228 – 231: They state: "Four occur in exons, of which three result in nonconservative amino-acid substitutions and one (rs1800404) encodes a synonymous substitution (in exon 10 of OCA2) and is located in a conserved binding site for transcription factor YY1 (known to regulate pigmentation in animal models⁵²)." They should cite and discuss the results in Crawford et al. 2017 which showed that the allele associated with light skin color at rs1800404 is associated with a shorter OCA2 gene transcript which is missing exon 10 and codes for a protein missing a transmembrane region.

Lines 255 – 257: "The 10q26 region newly associated with skin pigmentation shows SNPs with genome wide significant association spanning ~100Kb within an intergenic region of ~400Kb and

showing relatively low LD (Figure 5).” Did they mean to say low LD or high LD?

Lines 264 – 269: They mention *EMX2* as a candidate gene of interest which they propose may be regulated by nearby SNPs in association with skin pigmentation (though they lack any functional genomics data). For this candidate gene, and others, have they checked ENCODE or dbREGULOME databases that describe chromatin interactions to see if candidate regulatory SNPs are in regions interacting with this gene (or other candidate genes)?

Lines 270 – 285 and 356 - 358: The results at *MFSD12* are of particular interest given the recent characterization of this gene as playing a role in pigmentation in the study by Crawford et al. 2017. The identification of a non-synonymous SNP which is common only in Asians and Native Americans and shows a correlation with UV is of considerable interest because it may indicate evidence for convergent evolution. They mention that SNPs in this gene associated with pigmentation, as described in Crawford et al. 2017, are only variable in Africa. But in that study, they also identified a regulatory region upstream of *MFSD12* which is highly variable globally and should be mentioned in their text. They also state that the Crawford et al. 2017 paper showed that this gene plays a role in lysosomal biology using animal studies but they neglect to mention that those animal models also demonstrated a clear role for this gene in altering pigmentation—influencing both pheomelanin and eumelanin levels in vitro and in vivo. That study also demonstrated that *MFSD12* is expressed nearly 100 fold higher in melanocytes relative to other cell types which should be cited. Their observation that this protein is present in melanocytes in human skin cells is consistent with that prior study. Their observation that *MFSD12* is not present in the hair bulb is interesting.

Lines 304 – 312: The description of results of scans of selection is not adequate. They state: “Several studies have detected signatures of selection around many pigmentation genes. In agreement with those previous analyses we found strong signals of selection, in the 1000 Genomes (1KG) data, at most of the pigmentation-associated regions replicated here (Supplementary Figure 13 and Supplementary Table 7).” However, the results shown in the sup tables/figures show that for a number of the SNPs associated with pigmentation in their study there are not strong signals of selection. Furthermore, using an empirical threshold of “ $P < .05$ ”, meaning that their test statistics are in the extreme 5% of the empirical distribution, is not very stringent. The majority of genome wide studies of selection using empirical thresholds of 1% or less (many are as stringent as .01%). Based on the results shown in the sup docs, it appears that this 5% threshold may not be stringent enough and is likely to contain many false positives.

Figure 7A: The results of the PBS statistic are not very convincing for their candidate SNP. As stated above, a threshold of 5% is not very stringent. There are many SNPs that are outliers in this region extending across many genes despite high recombination in the region. I wonder how many more there would be if they zoomed out even further.

Reviewer #3 (Remarks to the Author):

The authors greatly clarified the rationale for their analytic strategy both in their response and in the revised manuscript. However, it should be stated up-front in the abstract that the reported results are from a conditional analysis, as it is unconventional. I would recommend changing the start of the

second sentence from "We found eighteen independent signals..." to "Conditional analysis identified eighteen independent signals...".

While they have corrected the majority of the grammatical errors, a few remain.

Finally, I maintain that the tables should report the p-values and not the log transformed p-values. Log transformed values are used for visual displays and I cannot find even a single example in the GWAS literature of a table that reports log transformed p-values rather than p-values. Finally, on-line databases perform data extraction from manuscript tables and GWAS catalogues report p-values, NOT log-transformed p-values (<https://www.ebi.ac.uk/gwas/search>). Unconventionally publishing transformed p-values requires either the curators or people downloading the database to perform an extra step of conversion so that results are comparable across studies.

Dear Editor

Many thanks for your email informing us of your decision regarding our manuscript “A GENOME-WIDE ASSOCIATION SCAN IN LATIN AMERICANS HIGHLIGHTS THE CONVERGENT EVOLUTION OF LIGHTER SKIN PIGMENTATION IN EURASIA”. We have now revised the manuscript as requested and provide a description of these changes below.

Regarding the conditional analyses. We now refer to the standard (unconditional) GWAS as the primary results. These are summarized in the new Table 1 and the new Manhattan plot. We describe the approach for the conditional analyses (and justification) briefly in the Methods, but these results are presented as secondary in the abstract, introduction and results, with the p-values moved to supplement. We hope to have streamlined presentation of the unconditional and conditional results as to avoid jumping back and forth between them. We trust that these changes have added clarity to the presentation of our results. Other than the changes to Table 1 and the Manhattan and LocusZoom plots, sections of the text with extensive revisions have been highlighted in green.

Regarding our presentation of $-\log(P\text{-values})$: we have now replaced these by P-values throughout the paper.

Regarding the other minor comments made by reviewer #2, we have incorporated them in the revised version of the manuscript as detailed below and highlighted (in yellow) in the manuscript:

Lines 228 – 231: They state: “Four occur in exons, of which three result in nonconservative amino-acid substitutions and one (rs1800404) encodes a synonymous substitution (in exon 10 of OCA2) and is located in a conserved binding site for transcription factor YY1 (known to regulate pigmentation in animal models52).” They should cite and discuss the results in Crawford et al. 2017 which showed that the allele associated with light skin color at rs1800404 is associated with a shorter OCA2 gene transcript which is missing exon 10 and codes for a protein missing a transmembrane region.

We thank the reviewer for pointing this out. We have included the additional information regarding SNP rs1800404 in the main text.

Lines 255 – 257: “The 10q26 region newly associated with skin pigmentation shows SNPs with genome wide significant association spanning ~100Kb within an intergenic region of ~400Kb and showing relatively low LD (Figure 5).” Did they mean to say low LD or high LD?

As shown in the amount of local recombination rate and the pairwise LD heatmap at 10q26, the region surrounding the associated SNP shows low LD. In order to avoid ambiguity we have deleted the word “relatively” from this sentence.

Lines 264 – 269: They mention EMX2 as a candidate gene of interest which they propose may be regulated by nearby SNPs in association with skin pigmentation (though they lack any functional genomics data). For this candidate gene, and others, have they checked ENCODE or dbREGULOME databases that describe chromatin interactions to see if candidate regulatory SNPs are in regions interacting with this gene (or other candidate genes)?

We have checked these databases but were unable to find any strong evidence. Following the previous advice of this reviewer we rather be conservative in our reference to these bioinformatics analyses, particularly since as correctly pointed out by the reviewer, they do not provide proof of a causal link explaining results.

Lines 270 – 285 and 356 - 358: The results at MFSD12 are of particular interest given the recent characterization of this gene as playing a role in pigmentation in the study by Crawford et al. 2017. The identification of a non-synonymous SNP which is common only in Asians and Native Americans and shows a correlation with UV is of considerable interest because it may indicate evidence for convergent evolution. They mention that SNPs in this gene associated with pigmentation, as described in Crawford et al. 2017, are only variable in Africa. But in that study, they also identified a regulatory region upstream of MFSD12 which is highly variable globally and should be mentioned in their text. They also state that the Crawford et al. 2017 paper showed that this gene plays a role in lysosomal biology using animal studies but they neglect to mention that those animal models also demonstrated a clear role for this gene in altering pigmentation—influencing both pheomelanin and eumelanin levels in vitro and in vivo. That study also demonstrated that MFSD12 is expressed nearly 100 fold higher in melanocytes relative to other cell types which should be cited. Their observation that this protein is present in melanocytes in human skin cells is consistent with that prior study. Their observation that MFSD12 is not present in the hair bulb is interesting.

We thank the reviewer for pointing this out. We have included the additional information mentioned by the reviewer in the main text of the manuscript.

Lines 304 – 312: The description of results of scans of selection is not adequate. They state: “Several studies have detected signatures of selection around many pigmentation genes. In agreement with those previous analyses we found strong signals of selection, in the 1000 Genomes (1KG) data, at most of the pigmentation-associated regions replicated here (Supplementary Figure 13 and Supplementary Table 7).” However, the results shown in the sup tables/figures show that for a number of the SNPs associated with pigmentation in their study there are not strong signals of selection. Furthermore, using an empirical threshold of “ $P < .05$ ”, meaning that their test statistics are in the extreme 5% of the empirical distribution, is not very stringent. The majority of genome wide studies of selection using empirical thresholds of 1% or less (many are as stringent as .01%). Based on the results shown in the sup docs, it appears that this 5% threshold may not be stringent enough and is likely to contain many false positives.

We agree with the reviewer that using an empirical threshold of P-value <0.05 might be too lenient. We have therefore decided to use a more stringent threshold of P-value <0.01 as suggested by the reviewer. The Supplementary Figures and Tables have been updated accordingly.

We also agree with the reviewer that for many of our associated variants there are no strong signals of selection. We are aware that for many of our novel associated variants the functional nature of their association to pigmentation is uncertain, and it is possible that selection has acted on other nearby SNPs. However, as can be seen in the Supplementary Figure 13, for 8 out of the 12 genomic regions that we report in Table 1, there is at least one selection statistic showing scores above the top 1% empirical threshold. That is why we decided to mention that there is evidence of selection at the pigmentation-associated genomic regions and not at the associated variants itself, although some known pigmentation associated variants do show strong signals of selection. We have changed the text in the manuscript in order to make this

clearer. More generally, we note that this result is also in agreement with our enrichment analysis presented in Supplementary Table 8.

Figure 7A: The results of the PBS statistic are not very convincing for their candidate SNP. As stated above, a threshold of 5% is not very stringent. There are many SNPs that our outliers in this region extending across many genes despite high recombination in the region. I wonder how many more there would be if they zoomed out even further.

We have updated Figure 7A showing the more stringent top 1% empirical threshold. Although the associated variant does not show a PBS score above this more stringent threshold, (it has however and empirical P-value equal to 0.0137), we note that the evidence for selection at this region should be considered in the light of our complementary analysis, including the correlation with solar radiation and the ABC analysis. As mentioned above, we do not intent to say that the associated variant has been the target of selection.

Reviewers' comments:

Reviewer #2 (Remarks to the Author):

In this resubmission the authors have addressed some, but not all, of my prior concerns.

They have done a good job presenting standard p values for the association results in the text and in Table 1. However, they have continued to highlight results for "conditional analyses" in the abstract, introduction, results, and conclusion, which is still confusing, particularly since this is a new approach that has not been independently peer reviewed (though at least now the conditional analysis results are better distinguished from the unconditioned results).

I found this particularly confusing: "As expected from the gain of power provided by conditioning on well-established pigmentation loci with large effects, P-values from the conditioned analyses (Supplementary Table 5) are smaller than those obtained in the unconditioned analyses (Table 1), including P-values for previously-reported pigmentation-associated SNPs not used in conditioning (rs1042602 in TYR, rs885479 in MC1R; Table 1, Supplementary Table 5)."

I am also concerned that they choose to focus subsequent analyses on the conditional GWAS results. Their justification is not clear to me: "Given the consistency of results from the unconditioned and conditioned analyses, for simplicity of presentation in what follows we focus mainly on the conditioned analyses, which are more informative because they remove variation due to other known genetic variants."

I am still confused by this sentence: "The 10q26 region that is newly associated with skin pigmentation shows SNPs with genome-wide significant association spanning ~100Kb within a low-LD intergenic region of ~400Kb (Figure 5)"

If there is low LD then presumably these multiple SNPs are independent? If so, that should be shown by standard conditional analysis and clearly stated.

They've done a good job citing a previously published study on MFSD12 demonstrating that it plays a role in pigmentation (Crawford et al. 2018). However, the following sentences are incorrect: "Variants upstream of MFSD12 are highly variable across human populations, but the SNPs recently associated with skin pigmentation in Sub-Saharan Africans are polymorphic mainly in Africa⁷." Line 372: "Strongest association was detected for SNPs that are polymorphic almost exclusively in African populations⁷."

There were two independent and equally strong and significant associations identified at MFSD12 in Africans. One was within the gene and was variable only in Africans and the second was in an upstream region which is highly polymorphic in global populations. It is incorrect to state that the SNPs associated with pigmentation near MFSD12 are only variable in Africa.

I do not see "strong signals of selection" at the candidate loci in the 1000 genomes samples. In the sup doc they see 10 signals that are in the extremes of the empirical distribution out of around 83 tests performed. How many signals would they expect to find by chance in that many tests. When they state that many of the strongest associations are nearby, how far are they? How do they know that they are due to selection acting on pigmentation as opposed to other loci in the regions? I'm also confused by the description of the analysis looking for enrichment of selection using the PBS statistic. They state that they are looking at candidate genes (+/- 2kb) rather than the candidate SNPs identified in the study. If the functional variants are often in non-coding regions and likely regulatory in nature, why would one even expect an enrichment in the genes themselves? Also, why not look at

enrichment for iHS results? They say they have more sites included in the PBS analysis (8 million) but they also have 3million sites included in the iHS statistic.

Line 526: What are "best guess imputed genotypes"?

Line 539: As part of their justification for using conditional analyses they state "The situation in case-control studies of disease is more complex because in that setting association testing is affected by disease prevalence and effect sizes,87,88 so that disease GWAS have only occasionally conditioned on established loci 89." Yet they also say that pigmentation is highly complex. It is also influence by "prevalence" of the phenotype and alleles and effect sizes. I am not getting this argument and I still don't think the use of the conditional analysis is well justified. The entire thing could be described in the supp text rather than highlighted repeatedly throughout the main text.

Line 601: They say in their response to reviewers that they looked at the top 1% of the empirical distribution though that is not explicitly stated in the methods (but should be).

I do not understand why they have a separate Table 1B that lists: Table 1B. Additional index SNPs in well-established genomic regions associated with pigmentation traits†.

Why not list ALL the results in a single Table. Table 1A also includes SNPs in "well-established genomic regions associated with pigmentation results". Or are these the conditional analyses? If so, they should not be listed in Table 1 which adds confusion.

On a minor note, why are the headers listed as A), B), C), etc?

Reviewers' comments:

Reviewer #2 (Remarks to the Author):

In this resubmission the authors have addressed some, but not all, of my prior concerns.

1. They have done a good job presenting standard p values for the association results in the text and in Table 1. However, they have continued to highlight results for “conditional analyses” in the abstract, introduction, results, and conclusion, which is still confusing, particularly since this is a new approach that has not been independently peer reviewed (though at least now the conditional analysis results are better distinguished from the unconditioned results).

We have revised the manuscript to remove mention of the conditional analysis from the abstract, cut down the mention of conditional analysis in the results (also see response to point 3 below), and removed the corresponding supplementary note. We also clarified in the introduction as well as results that it is a follow-up analysis. As with all supplementary analysis, the conditional analysis is mentioned in relevant places in the main text, but we have tried not to ‘highlight’ it.

There was another application of conditional analysis done in the results, where in each associated region we conditioned on the index SNP to see if any association still remained. In other words, this verifies if the index SNP is the only locus driving association in the region. This is a common check and is what the reviewer suggests in point 4. So we have not reduced the mention of this analysis in results.

2. I found this particularly confusing: “As expected from the gain of power provided by conditioning on well-established pigmentation loci with large effects, P-values from the conditioned analyses (Supplementary Table 5) are smaller than those obtained in the unconditioned analyses (Table 1), including P-values for previously-reported pigmentation-associated SNPs not used in conditioning (rs1042602 in TYR, rs885479 in MC1R; Table 1, Supplementary Table 5).”

We have modified the text to make it clearer.

3. I am also concerned that they choose to focus subsequent analyses on the conditional GWAS results. Their justification is not clear to me: “Given the consistency of results from

the unconditioned and conditioned analyses, for simplicity of presentation in what follows we focus mainly on the conditioned analyses, which are more informative because they remove variation due to other known genetic variants.”

We apologize for this confusion. Even though the last version of the manuscript was revised to make the unconditional analysis its main result, this statement inadvertently remained. We have now deleted this statement.

4. I am still confused by this sentence: “The 10q26 region that is newly associated with skin pigmentation shows SNPs with genome-wide significant association spanning ~100Kb within a low-LD intergenic region of ~400Kb (Figure 5)”. If there is low LD then presumably these multiple SNPs are independent? If so, that should be shown by standard conditional analysis and clearly stated.

The reviewer is correct in pointing out that this statement was confusing. We have rewritten the sentence to make it clear that the associated region is a single block of LD.

As the reviewer suggests here, we did check that there is only a single locus driving the association, by re-running the GWAS in the region conditioning on the top SNP. We mention this check briefly in the results, which was done in each associated region.

5. They’ve done a good job citing a previously published study on MFSD12 demonstrating that it plays a role in pigmentation (Crawford et al. 2018). However, the following sentences are incorrect: “Variants upstream of MFSD12 are highly variable across human populations, but the SNPs recently associated with skin pigmentation in Sub-Saharan Africans are polymorphic mainly in Africa7.” Line 372: “Strongest association was detected for SNPs that are polymorphic almost exclusively in African populations7.” There were two independent and equally strong and significant associations identified at MFSD12 in Africans. One was within the gene and was variable only in Africans and the second was in an upstream region which is highly polymorphic in global populations. It is incorrect to state that the SNPs associated with pigmentation near MFSD12 are only variable in Africa.

The reviewer is correct. We have altered the statement accordingly.

6A. I do not see “strong signals of selection” at the candidate loci in the 1000 genomes samples. In the sup doc they see 10 signals that are in the extremes of the empirical distribution out of around 83 tests performed. How many signals would they expect to find by chance in that many tests.

As suggested by the reviewers in the previous revision, we are now using the more stringent top 1% empirical threshold (i.e. empirical P-value <0.01). We note that these P-values were estimated using an outlier approach by ranking all the genome-wide scores and diving by the number of values in the distribution, taking the upper tail for PBS and iHS and the lower tail for Tajima’s D. It is therefore incorrect to say that we only “see 10 signals that are in the extremes of the empirical distribution of around 83 tests performed”. We have modified the methods section describing this analysis to make this clearer.

6B. When they state that many of the strongest associations are nearby, how far are they?

On Supplementary Figure 13 we show the distribution of selection statistics surrounding the genome-wide associated SNPs. This figure also includes a panel

showing the genes at the specific region with genomic coordinates. We now referred to this Supplementary Figure on the text when making this statement.

6C. How do they know that they are due to selection acting on pigmentation as opposed to other loci in the regions?

We agree with the reviewer that we cannot conclusively know whether the signals of selection found at the associated regions are due to selection acting on pigmentation phenotypes or other traits. However, we note that in order to investigate pleiotropic effects at our associated regions we would have to conduct several additional analyses, which is beyond the scope of this study.

6D. I'm also confused by the description of the analysis looking for enrichment of selection using the PBS statistic. They state that they are looking at candidate genes (+/- 2kb) rather than the candidate SNPs identified in the study. If the functional variants are often in non-coding regions and likely regulatory in nature, why would one even expect an enrichment in the genes themselves?

We agree with the reviewer that restricting our enrichment analysis to gene regions might not be the best strategy to test for enrichment of selection signals, as many of the associated variants might be present on intergenic regions. We have therefore modified our enrichment analysis to test whether the distribution of the PBS scores at haplotype blocks with associated SNPs (i.e. those with association P -value $<10^{-5}$) is significantly higher than the distribution of the PBS scores using the haplotype blocks in the rest of the genome. We note that in line with our previous enrichment analysis, this new analysis revealed a significant enrichment of selection PBS scores for all our pigmentation phenotypes.

6E. Also, why not look at enrichment for iHS results? They say they have more sites included in the PBS analysis (8 million) but they also have 3million sites included in the iHS statistic.

We have now extended our enrichment analysis to include iHS selection scores.

7. Line 526: What are "best guess imputed genotypes"?

We have added a clarification in the methods section.

8. Line 539: As part of their justification for using conditional analyses they state "The situation in case-control studies of disease is more complex because in that setting association testing is affected by disease prevalence and effect sizes,87,88 so that disease GWAS have only occasionally conditioned on established loci 89." Yet they also say that pigmentation is highly complex. It is also influence by "prevalence" of the phenotype and alleles and effect sizes. I am not getting this argument and I still don't think the use of the conditional analysis is well justified. The entire thing could be described in the supp text rather than highlighted repeatedly throughout the main text.

Following the reviewer's suggestion, we have further cut down the mention of conditional analysis, as explained in responses to points 1 and 3 above. The results of the conditional analysis are only presented in supplement, following the reviewer's suggestion. As with all supplementary analysis, they are mentioned in relevant places in the main text, but we have tried not to 'highlight' it.

The major difference we wanted to emphasize in that section is that pigmentation being a quantitative trait is quite different from binary traits used in case-control analysis. For example, the regression models being used are different: linear regression for quantitative traits and logistic regression for binary traits. The situation of binary traits in context of conditioning has been studied in detail in Pirinen et al. (2012), whose findings we mention in that paragraph. The applicability of conditioning in binary traits, as we mention following the paper, depends on the 'disease prevalence' i.e. the frequency of the binary trait. This concept of 'disease prevalence' or 'trait frequency' is not applicable to quantitative traits, and is not related to 'allele frequency' which is the property of a SNP, not a trait.

9. Line 601: They say in their response to reviewers that they looked at the top 1% of the empirical distribution though that is not explicitly stated in the methods (but should be).

We thank the reviewer for pointing this out. The methods section describing this analysis has now been modified.

10. I do not understand why they have a separate Table 1B that lists: Table 1B. Additional index SNPs in well-established genomic regions associated with pigmentation traits†. Why not list ALL the results in a single Table. Table 1A also includes SNPs in "well-established genomic regions associated with pigmentation results". Or are these the conditional analyses? If so, they should not be listed in Table 1 which adds confusion.

We agree with the reviewer and have converted them into a single table, all of them being unconditional results.

11. On a minor note, why are the headers listed as A), B), C), etc?

The alphabetical listing of the headers in the results section has now been removed.

REVIEWERS' COMMENTS:

Reviewer #2 (Remarks to the Author):

The authors have done an excellent job addressing the majority of concerns. I see just one section that needs editing.

They state: "In addition to allelic heterogeneity at these two genes here we identify rs224071 at MFSD12 as another pigmentation variant specific to populations of East Asian/Native American ancestry. This same gene has been recently implicated in a study of skin pigmentation variation in Sub-Saharan Africans. Strongest association was detected for SNPs that are polymorphic primarily in African populations but not seen in Europeans or East Asians. By contrast, we found that in our sample the strongest association with skin pigmentation is seen for the Y182H amino-acid substitution in MFSD12, a variant seen at high frequency only in East Asians and Native Americans."

As noted in my previous comments, there are strong statistically significantly associated SNPs with p values from $\sim 10e16$ to $\sim 10e18$ at two regions at the MFSD12 locus. One is within the gene and variable only in Africans and the other is upstream of the gene and variable in global populations. So, the following would be correct:

"In addition to allelic heterogeneity at these two genes here we identify rs224071 at MFSD12 as another pigmentation variant specific to populations of East Asian/Native American ancestry. This same gene has been recently implicated in a study of skin pigmentation variation in Sub-Saharan Africans. The strongest associations in that study were within synonymous and intronic regions of MFSD12, variable only in Africans, and in an upstream regulatory region, variable in global populations. By contrast, we found that in our sample the strongest association with skin pigmentation is seen for the Y182H amino-acid substitution in MFSD12, a variant seen at high frequency only in East Asians and Native Americans."

They might want to note that this is a nice example of convergent evolution.

Responses to Reviewer's comments (responses in bold):

Reviewer #2 (Remarks to the Author):

The authors have done an excellent job addressing the majority of concerns. I see just one section that needs editing.

They state: "In addition to allelic heterogeneity at these two genes here we identify rs224071 at MFSD12 as another pigmentation variant specific to populations of East Asian/Native American ancestry. This same gene has been recently implicated in a study of skin pigmentation variation in Sub-Saharan Africans. Strongest association was detected for SNPs that are polymorphic primarily in African populations but not seen in Europeans or East Asians. By contrast, we found that in our sample the strongest association with skin pigmentation is seen for the Y182H amino-acid substitution in MFSD12, a variant seen at high frequency only in East Asians and Native Americans."

As noted in my previous comments, there are strong statistically significantly associated SNPs with p values from $\sim 10^{-16}$ to $\sim 10^{-18}$ at two regions at the MFSD12 locus. One is within the gene and variable only in Africans and the other is upstream of the gene and variable in global populations. So, the following would be correct:

"In addition to allelic heterogeneity at these two genes here we identify rs224071 at MFSD12 as another pigmentation variant specific to populations of East Asian/Native American ancestry. This same gene has been recently implicated in a study of skin pigmentation variation in Sub-Saharan Africans. The strongest associations in that study were within synonymous and intronic regions of MFSD12, variable only in Africans, and in an upstream regulatory region, variable in global populations. By contrast, we found that in our sample the strongest association with skin pigmentation is seen for the Y182H amino-acid substitution in MFSD12, a variant seen at high frequency only in East Asians and Native Americans."

They might want to note that this is a nice example of convergent evolution.

We have edited this section as requested by the reviewer.